# PDDFORMER: PAIRWISE DISTANCE DISTRIBUTION GRAPH TRANSFORMER FOR CRYSTAL MATERIAL PROPERTY PREDICTION

## ABSTRACT

The crystal structure can be simplified as a periodic point set repeating across the entire three-dimensional space along an underlying lattice. Traditionally, methods for representing crystals rely on descriptors like lattice parameters, symmetry, and space groups to characterize the structure. However, in reality, atoms in material always vibrate above absolute zero, causing continuous fluctuations in their positions. This dynamic behavior disrupts the underlying periodicity of the lattice, making crystal graphs based on static lattice parameters and conventional descriptors discontinuous under even slight perturbations. To this end, chemists proposed the Pairwise Distance Distribution (PDD) method, which has been used to distinguish all periodic structures in the world's largest real materials collection, the Cambridge Structural Database. However, achieving the completeness of PDD requires defining a large number of neighboring atoms, resulting in high computational costs. Moreover, it does not account for atomic information, making it challenging to directly apply PDD to crystal material property prediction tasks. To address these challenges, we propose the atom-Weighted Pairwise Distance Distribution (WPDD) and Unit cell Pairwise Distance Distribution (UPDD) for the first time, incorporating them into the construction of multi-edge crystal graphs. Based on this, we further developed WPDDFormer and UPDDFormer, graph transformer architecture constructed using WPDD and UPDD crystal graphs. We demonstrate that this method maintains the continuity and completeness of crystal graphs even under slight perturbations in atomic positions. Moreover, by modeling PDD as global information and integrating it into matrix-based message passing, we significantly reduced computational costs. Comprehensive evaluation results show that WPDDFormer achieves state-of-the-art predictive accuracy across tasks on benchmark datasets such as the Materials Project and JARVIS-DFT.

## 1 INTRODUCTION

Crystals are solids with a regular geometric shape formed by atoms, ions, or molecules arranged periodically in space during the crystallization process. Their structure is typically described using repeating unit cells and lattice vectors. However, this method of description brings a fundamental challenge: the same crystal structure can be represented by different unit cells and lattice vectors, as shown in Figure 1a. Additionally, in real-world scenarios, the experimental coordinates of unit cells and atoms are inevitably affected by atomic vibrations and measurement noise. These subtle disturbances can lead to discontinuous changes in any simplified unit cell (Kurlin, 2024), resulting in numerous different unit cells for a given crystal structure, as shown in Figure 1b, thereby introducing ambiguity in the representation of crystal data (Widdowson & Kurlin, 2022). Currently, many graph neural networks (Batzner et al., 2022; Yan et al., 2022; 2024a;b) typically use unit cell parameters, simplified cell parameters, symmetry, and space groups to represent the periodic structure of crystals. However, these features are either non-invariant or discontinuous (Zwart et al., 2008) invariants, leaving the issue of ambiguity in crystal data unresolved (Patterson, 1944; Widdowson et al., 2022; Groom et al., 2016; Bartók et al., 2013; Wassermann et al., 2010; Ahmad et al., 2018).

The continuous and complete invariant—Pairwise Distance Distribution (PDD)—proposed by Widdowson and Kurlin (2022) addresses the ambiguity in crystal data representation by distinguish-

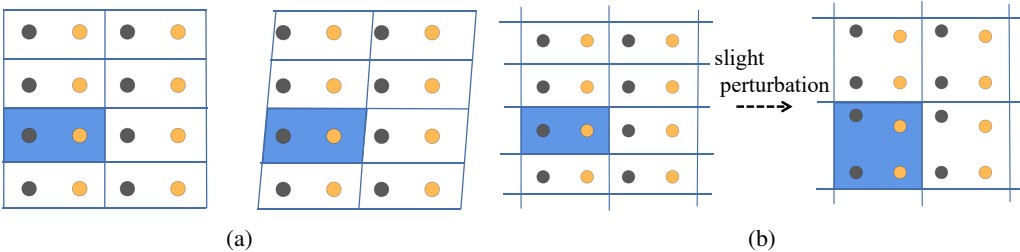

Figure 1: Illustrations of different unit cell and lattice representations of the same crystal structure. The blue area in the figure represents possible unit cell structures. Figure (a) shows several possible choices among the infinitely many unit cells for the same crystal structure in the undisturbed case. Figure (b) illustrates that for almost any perturbation, the symmetry group and any reduced unit cell (with minimal volume) will undergo discontinuous changes.

ing all periodic structures in the world's largest real material collection, the Cambridge Structural Database. To achieve completeness, PDD requires a predetermined number of sufficient neighbors, which is computationally expensive and difficult to directly apply for predicting crystal properties (Balasingham et al., 2022). Balasingham et al. (2024) employed distance distribution graphs (DDGs) based on PDD to predict the properties of crystal materials, but they did not achieve satisfactory performance (only slightly better than CGCNN), and although this approach reduced computational costs, it compromised the completeness of PDD. In contrast, crystal graph representations based on multi-edge crystal graphs and unit cell parameters (Taniai et al., 2024; Yan et al., 2024a) achieve completeness, more accurately characterizing crystal structures and achieving state-of-the-art performance in crystal material property prediction tasks. However, the use of unit cell parameters leads to discontinuities in the crystal graphs.

Since PDD does not account for atomic types, it is challenging to use it directly for effective crystal property prediction. To better represent crystal structures, we first introduce atom-Weighted PDD (WPDD) and intra-Unit cell PDD (UPDD). Furthermore, we integrate WPDD and UPDD into the construction of multi-edge crystal graphs and propose the PDD Graph Transformer (including WPDDFormer and UPDDFormer) based on the transformer architecture. We model WPDD as global information and incorporate it into matrix-based message passing, significantly reducing computational costs (as shown in Table 3). Finally, we employ the Earth Mover's Distance (EMD) (Rubner et al., 2000) to assess the continuity of crystal graphs, demonstrating that WPDD crystal graphs constructed using only Euclidean distances maintain continuity and general completeness[1] under slight atomic position perturbations, providing a more accurate depiction of real crystal structures. Ablation experiments show the crucial role of (W/U)PDD in constructing crystal graphs. Through comprehensive evaluations, our method achieves state-of-the-art predictive accuracy across various tasks in the Materials Project (Chen et al., 2019) and JARVIS-DFT (Choudhary et al., 2020) datasets. This advancement highlights the effectiveness of WPDDFormer in bridging the gap between traditional crystal descriptors and dynamic atomic behavior, leading to more accurate and reliable predictions in materials science.

## 2  PRELIMINARIES

In this section, we introduce the definitions of crystal structures, PDD, isometric crystal graphs, and the continuity and geometric completeness of crystal graphs. Additionally, we provide in Appendix A.4 the definition and proof of the unique geometric constraints of crystals.

### 2.1  THE STRUCTURE OF CRYSTALS

By selecting an appropriate structural unit, the entire crystal structure can be viewed as the periodic repetition of this unit in space. This property, where atoms within a crystal repeat in three-dimensional space according to a specific pattern, is called periodicity, with the smallest repeatable structural unit being the unit cell. The unit cell can be defined as $\mathcal{U} = (\mathcal{X}, \mathcal{P})$, where $\mathcal{X}$

---

[1]Except for chiral crystal structures and theoretically extremely large unit cells.

and $\mathcal{Z}$ can be represented in matrix form. Typically, $\mathcal{X} = [x_1, x_2 \cdots x_{n-1}, x_n]^T \in \mathbb{R}^{n \times 1}$, where $n$ represents the number of atoms and $x_i \in \mathbb{R}^1$ represents the atomic type of atom $i$ in the unit cell. $\mathcal{P} = [p_1, p_2 \cdots p_{n-1}, p_n]^T \in \mathbb{R}^{n \times 3}$ is the atomic position matrix, where $p_i \in \mathbb{R}^3$ represents the Cartesian coordinates of the atom $i$ in the unit cell in 3D space. The lattice vectors $\mathcal{L} = [l_1, l_2, l_3]^T \in \mathbb{R}^{3 \times 3}$ can reflect the way the unit cell repeats in three directions to map the periodic crystal structure. Therefore, in 3D space, the infinite crystal structure $\mathcal{S}$ can be represented as $(\mathcal{U}, \mathcal{L})$.

## 2.2 Continuity and geometric completeness of crystal graphs

**Definition 1: Pointwise Distance Distribution.** For the infinite crystal structure $\mathcal{S} = (\mathcal{U}, \mathcal{L})$ mentioned in Section 2.1, fix a neighbor count $k \geq 1$. For each point $x_i$ in the unit cell $\mathcal{U}$, let $d_{i1} \leq \cdots \leq d_{ik}$ be the Euclidean distances from $\mathbf{p}_i$ to its $k$ nearest neighbors in the infinite crystal structure. Consider an $n \times k$ matrix composed of $n$ rows of distance vectors, where each point $x_i \in \mathcal{U}$ corresponds to one row. If the matrix contains $m \geq 1$ identical rows, they are merged into one row with a weight of $\frac{m}{n}$. The resulting matrix can be regarded as a weighted distribution of rows, which is called the Pointwise Distance Distribution $\mathcal{PDD}(S; k) \in \mathbb{R}^{n \times (k+1)}$.

**Definition 2: Isometric Crystal Graphs.** According to the definition from Widdowson & Kurlin (2022) and Yan et al. (2024a), an isometric transformation is a mapping that preserves Euclidean distances, denoted as $f(x) = Rx + b$. Any isometric transformation $f$ can be decomposed into translation, rotation, and reflection. Specifically, suppose there exists a rotation matrix $R \in \mathbb{R}^{3 \times 3}$, with a determinant of 1 ($|R| = 1$), and a translation vector $b \in \mathbb{R}^3$, then two crystal structures $\mathcal{S} = (\mathcal{U}, \mathcal{L})$ and $\mathcal{Q} = (\mathcal{U}', \mathcal{L}')$ are isometric, satisfying $\mathcal{U}' = R\mathcal{U} + b$, where $R\mathcal{U} + b$ denotes the application of the rotation $R$ and translation $b$ to each element in the infinite set $\mathcal{U}$.

If $\mathcal{S}$ and $\mathcal{Q}$ are isometric, then their crystal graph representations satisfy $\mathcal{G}(\mathcal{S}) = \mathcal{G}(\mathcal{Q})$, which means that the graphical representation of the crystal structure produces no false positives; that is, there are no isometric pairs where $\mathcal{G}(\mathcal{S}) \neq \mathcal{G}(\mathcal{Q})$ but $\mathcal{S} \simeq \mathcal{Q}$. Conversely, if $\mathcal{G}(\mathcal{S}) = \mathcal{G}(\mathcal{Q})$, then $\mathcal{S}$ and $\mathcal{Q}$ are isometric, meaning f produces no false negatives, i.e., there are no non-isometric pairs where $\mathcal{G}(\mathcal{S}) = \mathcal{G}(\mathcal{Q})$ but $\mathcal{S} \not\simeq \mathcal{Q}$. That is, if the crystal graph representations of artificially constructed crystal structures are identical under isometric transformations, then they are geometrically equivalent.

**Definition 3: Geometrically Complete Crystal Graphs.** According to Widdowson & Kurlin (2022) and Yan et al. (2024a), if we construct crystal graphs $\mathcal{G}(\mathcal{S}) = \mathcal{G}(\mathcal{Q}) \implies \mathcal{S} \simeq \mathcal{Q}$, where $\simeq$ denotes the isomorphism of two crystals as defined in Definition 2, then the crystal graph $\mathcal{G}$ is geometrically complete. This means that if two crystal graphs $\mathcal{G}(\mathcal{S})$ and $\mathcal{G}(\mathcal{Q})$ are identical, the infinite crystal structures represented by $\mathcal{G}(\mathcal{S})$ and $\mathcal{G}(\mathcal{Q})$ are also identical. If the constructed crystal graph $\mathcal{G}$ can distinguish any subtle structural differences between different crystal materials, it is said to be geometrically complete. According to Widdowson & Kurlin (2022), we present Definitions 4-6.

**Definition 4: Metric.** The metric $d$ between crystal graphs $\mathcal{G}$ satisfies all the axioms: 1) $d(\mathcal{G}(\mathcal{S}) = \mathcal{G}(\mathcal{Q})) = 0$ if and only if $\mathcal{G}(\mathcal{S}) = \mathcal{G}(\mathcal{Q})$; 2) Symmetry: $d(\mathcal{G}(\mathcal{S}), \mathcal{G}(\mathcal{Q})) = d(\mathcal{G}(\mathcal{Q}), \mathcal{G}(\mathcal{S}))$; 3) Triangle inequality: $d(\mathcal{G}(\mathcal{S}), \mathcal{G}(\mathcal{Q})) + d(\mathcal{G}(\mathcal{Q}), \mathcal{G}(\mathcal{K})) \geq d(\mathcal{G}(\mathcal{S}), \mathcal{G}(\mathcal{K}))$.

**Definition 5: Lipschitz continuity of crystal graphs.** If $\mathcal{Q}$ is obtained by moving each point in the periodic crystal $\mathcal{S} \subset \mathbb{R}^n$ by no more than $\epsilon$, and the distance of the constructed crystal graph structures satisfies $d(\mathcal{G}(\mathcal{S}), \mathcal{G}(\mathcal{Q})) \leq C\epsilon$, where $C$ is a constant, then the crystal graph is continuous, and $\mathcal{Q}, \mathcal{S} \subset \mathbb{R}^n$ can be any periodic crystal structures.

**Definition 6: EMD.** Let $\mathcal{G}(\mathcal{S})$ and $\mathcal{G}(\mathcal{Q})$ be the crystal graph structures we construct for periodic crystals $\mathcal{S}$ and $\mathcal{Q} \in \mathbb{R}^n$. The flow from $\mathcal{G}(\mathcal{S})$ to $\mathcal{G}(\mathcal{Q})$ is represented by an $n(\mathcal{S}) \times n(\mathcal{Q})$ matrix, where the elements $f_{ij} \in [0, 1]$ indicate the partial flow from $\mathcal{R}_i(\mathcal{S})$ to $\mathcal{R}_j(\mathcal{Q})$. The Earth Mover's Distance (EMD) is defined as the minimum cost: $EMD(\mathcal{G}(\mathcal{S}), \mathcal{G}(\mathcal{Q})) = \sum_{i=1}^n \sum_{j=1}^n f_{ij} |R_i(\mathcal{S}) - R_j(\mathcal{Q})|$ where $f_{ij} \in [0, 1]$ satisfies the following conditions:

$$\sum_{i=0}^n f_{ij} \leq w_i(\mathcal{S}), \ \sum_{j=0}^n f_{ij} \leq w_j(\mathcal{Q}), \ \sum_{i=1}^n \sum_{j=1}^n f_{ij} = 1 \qquad (1)$$

The first condition $\sum_{i=0}^n f_{ij} \leq w_i(\mathcal{S})$ means that not more than the weight $w_i(\mathcal{S})$ of the component $R_i(\mathcal{S})$ 'flows' into all components $R_j(\mathcal{Q})$ via 'flows' $f_{ij}$. Similarly, the second condition $\sum_{j=0}^n f_{ij} \leq w_j(\mathcal{Q})$ means that all 'flows' $f_{ij}$ from $R_i(\mathcal{S})$ 'flow' Into $R_j(\mathcal{Q})$ up to the maximum

weight $w_j(\mathcal{Q})$. The last condition $\sum_{i=1}^{n} \sum_{j=1}^{n} f_{ij} = 1$ forces to 'flow' all rows $R_i(\mathcal{S})$ to all rows $R_j(\mathcal{Q})$.

## 3 RELATED WORK

**Finite crystal graph representation.** CGCNN (Xie & Grossman, 2018) predicts material properties by learning the connections between atoms in crystals by representing crystal structures as finite multi-edge crystal graphs. Building on the construction of multi-edge crystal graphs, MegNet (Chen et al., 2019) introduced global state attributes into graph networks, while GATGNN (Louis et al., 2020) utilized multiple graph attention layers (GAT) to learn the properties of local neighborhoods and employed global attention layers to weight global atomic features. ALIGNN (Choudhary & De-Cost, 2021) and M3GNet (Chen & Ong, 2022) incorporated angular information into the message-passing process to generate richer and more discriminative representations. CrysMMNet (Das et al., 2023) adopted a multimodal framework, integrating graph and text representations to produce joint multimodal representations of crystalline materials. CrysDiff (Song et al., 2024) is a pretraining-finetuning framework based on diffusion models. However, the aforementioned methods represent crystals as finite graph structures, failing to capture the periodicity of infinite crystals effectively.

**Periodic representation of crystals.** Recently, Matformer (Yan et al., 2022) encoded periodic patterns by adding self-connecting edges to atoms based on lattice parameters, directly using lattice parameters to encode periodic structures under ideal conditions. PotNet (Lin et al., 2023) considered the infinite summation of interatomic interactions. Crystalformer (Taniai et al., 2024) performed infinite summations of interatomic potentials through infinitely connected attention while also utilizing lattice parameters. ComFormer (Yan et al., 2024a) constructed cell parameters by adding self-connecting edges to atoms and their copies in three different directions to encode periodic patterns, employing equivariant vector representations and invariant geometric descriptors of Euclidean distances and angles to represent the geometric information of crystals. GMTNet (Yan et al., 2024b) aims to predict the tensor properties of crystalline materials while satisfying O(3) group equivariance and the symmetry of crystal space groups to ensure the accuracy and consistency of tensor predictions. However, while these methods achieve complete crystal graph representations, the crystal structures they represent rely on non-invariants or discontinuous invariants, such as lattice parameters, symmetry, and space groups, failing to address the issue of crystal data fuzziness.

**Continuity and complete representations for crystals.** Addressing the continuity and completeness of crystal representations is a critical issue. Recent advancements in AMD (Wang et al., 2022) and PDD (Widdowson & Kurlin, 2022) have developed matrix forms that are both complete and continuous. However, in practical applications, using these matrix representations as inputs for predicting crystal properties without compromising continuity and completeness is challenging. The AMD and PDD representations are designed for stable crystal structures and do not account for atomic types. Their completeness relies on the assumption that no two crystals with identical structures differ solely by atomic type, which is only feasible for stable structures. Additionally, to achieve completeness, a sufficiently large number of neighbors $k$ must be predetermined for any test crystal. Typically, hundreds of neighbors are required to distinguish all periodic structures in the Cambridge Structural Database. Directly modeling PDD as edge information is impractical and costly in real-world applications (Balasingham et al., 2022).

## 4 PDDFORMER

In this section, we first propose two variants of PDD, namely WPDD and UPDD, and then incorporate them into crystal graph construction. We finally present the PDDformer framework.

### 4.1 WPDD

Since the PDD representation is designed for stable crystal structures and does not consider atomic types, it is not suitable for predicting crystal material properties. To account for the influence of atomic types, for a given crystal structure $\mathcal{S} = \mathcal{U} + \mathcal{L}$, where each atom $x_i \in \mathcal{U}$ is labeled with the atomic mass $t(x_i)$ corresponding to it, the final weight for each row is $\mathcal{W} = [w_1, \ldots, w_n]^T$, where $w_i = \frac{t(x_i)}{\sum_{j=1}^{n} t(x_j)}$. By concatenating this with PDD$\in \mathbb{R}^{n \times k}$, an atomic-mass-weighted

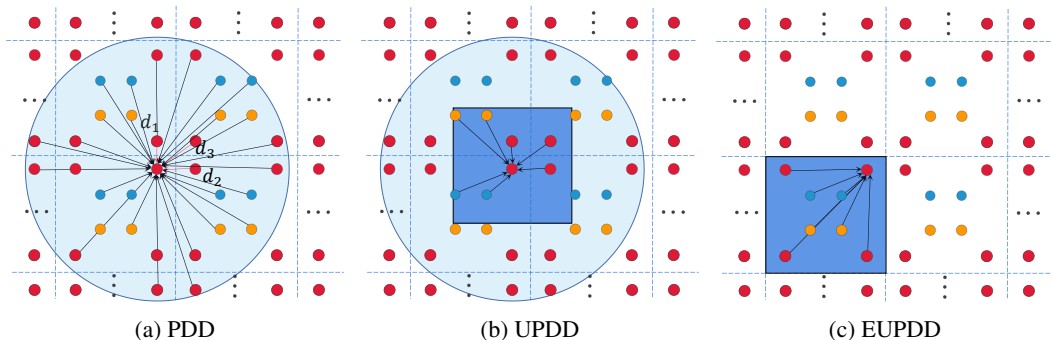

|                  |                   |                    |
|------------------|-------------------|--------------------|
| (a) PDD          | (b) UPDD          | (c) EUPDD          |

Figure 2: Schematic diagram of the selected neighbors in PDD in 2D. The edges in Figure a show the neighbor selection for atom $i$ in WPDD, represented by the red lines $d_1, d_2, d_3$. They are ordered by Euclidean distance as $d_1 < d_2 < d_3$. By comparing Figures (b) and (c), we can see that we construct the unit cell centered around each atom and select neighbors, rather than being limited to the unit cell where the atoms are located.

WPDD$(S; k) \in \mathbb{R}^{n \times (k+1)}$ is formed, represented by the following equation:

$$WPDD = (W, PDD) = \bigcup_{i=1}^n \left( \frac{t(x_i)}{\sum_{j=1}^n t(x_j)}, \bigcup_{j=1}^k \sqrt{(p_i - p_j)^2} \right) \tag{2}$$

Here, $n$ represents the number of atoms in the unit cell, and $p_i$ and $p_j$ denote the spatial coordinates of an atom $i$ and its neighbor $j$, respectively, and $k$ is the number of nearest neighbors selected when constructing the PDD, sorted in ascending order of Euclidean distance as $d_{i1} \leq \cdots \leq d_{ik}$, as shown in Figure 2a. WPDD is equivalent to the PDD of crystal structure $S$, except that the rows are not grouped as in the original version. This prevents the loss of atomic information when two primitive points have the same k-nearest neighbor distances but correspond to different atomic types. Therefore, WPDD$\in \mathbb{R}^{n \times (k+1)}$, where $n$ is the number of atoms in the constructed graph.

### 4.2 UPDD

When ensuring the completeness of PDD, a large number of neighbors must be predefined, typically requiring information on hundreds of neighbors, and in extreme cases, the number must exceed the atom count in any unit cell within the dataset. The number of neighbors, $k$, is difficult to determine across different datasets, and for unit cells with fewer atoms, which constitute a larger proportion of the dataset, an excess of neighbor information may interfere with the speed of message aggregation, leading to greater resource consumption.

To address this issue, we introduce Unit-cell PDD (UPDD). We achieve this by reconstructing the unit cell around each atom and encoding the pairwise distances between the atom and other atoms within the reconstructed unit cell. This means that when constructing PDD, we focus more on the overall structure of the atoms within the reconstructed unit cell, thereby reducing interference from excessive neighbor information. UPDD is defined by the following formula:

$$\mathcal{D}_i = \left\{ \bigcup_{j=1}^n \sqrt{(\overline{p_i} - \overline{p_j})^2} \mid i, j \in \mathcal{Z} \right\}, \ (\overline{p_i} - \overline{p_j})^2 = (\overline{x_i} - \overline{x_j})^2 + (\overline{y_i} - \overline{y_j})^2 + (\overline{z_i} - \overline{z_j})^2,$$
$$UPDD = \left\{ \bigcup_{i=1}^n \frac{1}{\mathcal{D}_i} \mid i \in \mathcal{Z}, \mathcal{D}_i \neq 0 \right\} \tag{3}$$

where $\mathcal{D}_i$ represents the union of feature vectors of distances between the atom $i$ and other atoms within the unit cell centered on the atom $i$, with $\mathcal{D}_i \in \mathbb{R}^n$, and $n$ representing the number of atoms in the unit cell. UPDD$\in \mathbb{R}^{n \times n}$ represents the union of distance features between all atoms. Since the interaction energy between an atom and its neighboring atoms is usually inversely proportional to the distance, we take the reciprocal of the distance feature after removing zeros.

As shown in Figure 2b, the selection is not based on Euclidean distances, but rather on choosing atoms within the reconstructed unit cell for construction. The dimension of our UPDD is determined by the atoms in the unit cell and does not require consideration of the neighbor count, k, across

different datasets, making it more generalizable. This UPDD covers unit cell structures with a larger number of atoms while also ensuring that unit cell structures with fewer atoms are not disturbed by excessive neighbor information. It also reduces resource consumption. Due to this crystal-specific treatment, the UPDD dimensions of different crystal structures may not match, so dimension alignment is required before feeding into the neural network.

## 4.3 CRYSTAL GRAPH CONSTRUCTION

By introducing PDD, we constructed a complete and continuous multi-edge crystal graph. In the graph, each node represents an atom $i$ and all its infinite duplicates in 3D space, with positions $\{\hat{p}_i | \hat{p}_i = p_i + k_1 l_1 + k_2 l_2 + k_3 l_3, k_1, k_2, k_3 \in \mathcal{Z}\}$, and node features $x_i$. An edge is established from node $j$ to node $i$ when the Euclidean distance $|e_{j'i}|^2$ between a duplicate of $j$ and $i$ satisfies $|e_{j'i}|^2 = |p_j + k_1' l_1 + k_2' l_2 + k_3' l_3 - p_i|^2 \leq r$, where $r \in \mathbb{R}$ is the cutoff radius. We select the nearest $t$ edges within the cutoff radius, each with a corresponding edge feature $|e_{j'i}|^2$. Since WPDD requires a large number of neighbors to be predefined, representing this neighbor information as edge features is neither practical nor cost-effective in real-world applications. Therefore, we retain its matrix form and incorporate it into the construction of the multi-edge crystal graph as a way to reflect the global information of the crystal structure. After passing through the Embedding Block in Section 4.6, UPDD is aligned in dimensions and transformed into matrix form data. Formally, we represent the constructed crystal graph as $\mathcal{G} = (\mathcal{X}, \mathcal{XI}, \mathcal{E}, PDD)$. Therein, $x_i \in \mathcal{X}$ is the feature vector of the atom $i$, $e_{ij}^h \in \mathcal{E}$ is the feature vector of the $h$-th edge between nodes $i$ and $j$, and we denote $\mathcal{XI}$ as the indices of the nodes $i$ and $j$ that form the edge. Sections 4.4 and 4.5 are our proofs of the continuity and geometric completeness of PDD crystal graphs.

## 4.4 CONTINUITY OF PROPOSED CRYSTAL GRAPHS

The continuity of the constructed crystal graph $\mathcal{G}(\mathcal{S})$ under perturbations of the crystal structure $\mathcal{S}$ will be measured using the EMD (Rubner et al., 2000), which applies to crystal graphs of any size. Definition 6 applies to any crystal graph $\mathcal{G}(\mathcal{S}) = ([w_1(S), R_1(S)], \ldots, [w_1(S), R_1(S)])$, where $[w_i(S), R_i(S)]$ represent the information extracted based on atom $i$ in the unit cell. $R_i(S) = R_i(\mathcal{S_X}, \mathcal{S_{XI}}, \mathcal{S_E}, \mathcal{S}_{PDD})$ includes atomic information, neighbor information used in constructing the multi-edge crystal graph, and the PDD invariants of the crystal structure $\mathcal{S}$, with weights $w_i \in (0, 1]$ satisfying the normalization condition $\sum_{i=1}^n w_i(\mathcal{S}) = 1$.

Subsequently, we only consider the case where the weighted vector $[w_i, R_i]$ corresponds to the $i$-th row of PDD($S; k$). Here, $n$ denotes the number of rows in PDD($S; k$). The size of each row $R_i(S)$ should be independent of $\mathcal{S}$ and depend solely on the number of neighbors $k$ in PDD($S; k$). For any vectors $R_i = (r_{i1}, \ldots, r_{ik})$ and $R_j = (r_{j1}, \ldots, r_{jk})$ of length $k$, we use the $L_\infty$- distance $|R_i - R_j|_\infty = \max_{l=0,\ldots k} |r_{il} - r_{jl}|_\infty$.

**Proposition 1.** The WPDD and UPDD multi-edge crystal graph is continuous.

**Proof:** For any $k \geq 1$, if the periodic crystal $\mathcal{S}, \mathcal{Q} \in \mathbb{R}^n$ satisfy $d_B(\mathcal{S}, \mathcal{Q}) < r(\mathcal{S})$, then we have: $EMD(\mathcal{G}(\mathcal{S}), \mathcal{G}(\mathcal{Q})) = EMD((\mathcal{S_X}, \mathcal{S_{XI}}, \mathcal{S_E}, \mathcal{S}_{PDD}), (\mathcal{Q_X}, \mathcal{Q_{XI}}, \mathcal{Q_E}, \mathcal{Q}_{PDD})) = EMD((\mathcal{S_X}, \mathcal{Q_X})) + EMD((\mathcal{S_{XI}}, \mathcal{Q_{XI}})) + EMD((\mathcal{S_E}, \mathcal{Q_E})) + EMD((\mathcal{S}_{PDD}, \mathcal{Q}_{PDD}))$. Since disturbances only change the positions of atoms and do not alter their types, therefore $EMD((\mathcal{S_X}, \mathcal{Q_X})) = 0$ and $EMD((\mathcal{S_{XI}}, \mathcal{Q_{XI}})) = 0$. So, we obtain $EMD(\mathcal{G}(\mathcal{S}), \mathcal{G}(\mathcal{Q})) = EMD((\mathcal{S_E}, \mathcal{Q_E})) + EMD((\mathcal{S}_{PDD}, \mathcal{Q}_{PDD})) \leq 2d_B(\mathcal{S}, \mathcal{Q})$.

The bottleneck distance $d_B(\mathcal{S}, \mathcal{Q}) < r(\mathcal{S})$ is defined as: $d_B(\mathcal{S}, \mathcal{Q}) = \inf_{g:\mathcal{S} \to \mathcal{Q}} \sup_{p \in \mathcal{S}} |p - g(p)|$ and the envelope radius $r(\mathcal{S})$ is the minimum half-distance between any two points in $r(\mathcal{S})$. In other words, $r(\mathcal{S})$ is the maximum radius of non-overlapping open balls centered at all points in $\mathcal{S}$. This implies that any small perturbation in atomic positions under the $d_B$ (Carstens et al., 1999) will lead to minor changes in the distribution of distances between points in the EMD.

Since the EMD between the constructed crystal graphs only relates to Euclidean distance. Euclidean distance itself is continuous, Theorem 1 extends the following fact: for a unit cell structure with two atoms, when the number of neighbors $k = 1$, if we perturb at most two points by $\epsilon$, the change in distance between the two points will be at most $2\epsilon$. Extending this to $n$ atomic points with $k$ neighbors, if we perturb at most $n$ points by $\epsilon$, the change in distance between $n$ points will be

at most $2nk\epsilon$. This aligns with Definition 5, hence the constructed WPDD and UPDD multi-edge crystal graph is continuous.

## 4.5 Geometric completeness of proposed crystal graphs

**Proposition 2.** The WPDD multi-edge crystal graph is geometrically complete.

Inspired by Yan et al. (2024a). We prove this by mathematical induction. Suppose the number of atoms (nodes) in the unit cell of a crystal is $n$.

**Base Case:** When $n = 1$, the infinite crystal structure represented by the WPDD multi-edge crystal graph is unique.

**Induction Hypothesis:** When $n \leq m$, the infinite crystal structure is unique.

**Induction Step:** Let $n = m + 1$. Without loss of generality, we safely assume that among the existing mmm nodes, $N_j$ is the set of nodes forming the local region for node $j$. Then, $j$ is the index of the $(m + 1)$-th node that is newly connected to these nodes. To prove that the infinite crystal structure remains unique, we only need to demonstrate that the relative position of node j is uniquely determined, given the WPDD multi-edge crystal graph. With this, the proposed WPDD multi-edge crystal graph can define a unique infinite crystal structure.

Here, we prove that the relative position of the newly added node $j$ is uniquely determined by the proposed WPDD multi-edge crystal graph.

**Proof:** We use proof by contradiction. First, assume that there exist two distinct relative positions $j$ and $j'$ that have the same WPDD multi-edge crystal graph, and we show that this assumption leads to a contradiction.

Since UPDD is constructed based on the size of the unit cell, when the number of atoms in the unit cell is relatively small, it could theoretically result in different crystal structures, where all atoms have the same Euclidean distances and atom types but inconsistent atomic positions (which do not exist in the real world), sharing the same crystal graph representation. According to the WPDD multi-edge crystal graph construction process described in Section 4.3, if two distinct crystal structures have the same WPDD crystal graph, their WPDD and the atomic types embedded by CGCNN must be identical. Since the WPDD crystal graph, which includes atomic information, is completely invariant, different crystal structures must have distinct WPDD crystal graphs, Relevant details can be found in Appendix 5. This contradicts the assumption. Hence, the proof is complete. Therefore, the proposed identical crystal graph can represent only the same infinite crystal structure. Then, based on Definition 3, we complete the proof of Proposition 2.

Finally, we conclude that the UPDD crystal graph can only guarantee continuity, while the WPDD crystal graph can ensure both continuity and completeness.

## 4.6 Network architecture

Based on the graph in Section 4.3, we propose the information propagation scheme of PDDFormer. The information propagation scheme of PDDFormer consists of four parts: the graph embedding Block, node-wise transformer Block (inspired by Yan et al. (2024a)) for details, refer to Appendix A.2, $PDD$ message passing Block, and output Block. Figure 3 illustrates the overall framework architecture of PDDFormer.

**Feature embedding block.** First, we introduce the construction of the graph embedding Block. We use atomic encoding from CGCNN for embedding. For the edge information $e_{ij}^h$, we employ radial basis functions to encode the distance between two adjacent nodes in the graph, represented by Equation 4, where $\gamma$ and $\mu$ are hyperparameters. For UPDD, due to the varying feature dimensions of UPDD for different crystals, we perform matrix multiplication on UPDD to align the structural information of different crystals, obtaining information for the PDD message passing layer. Thus, we obtain the graph embedding as:

$$\mathcal{A} = \mathcal{CGCNN}(\mathcal{X}), \; e_{ij}^h = \exp\left(-\gamma\left(\frac{\|p_i - p_j\|^2}{\mu}\right)\right), \; PDD = UPDD \otimes \mathcal{A} \qquad (4)$$

where $\otimes$ denotes the Hadamard product.

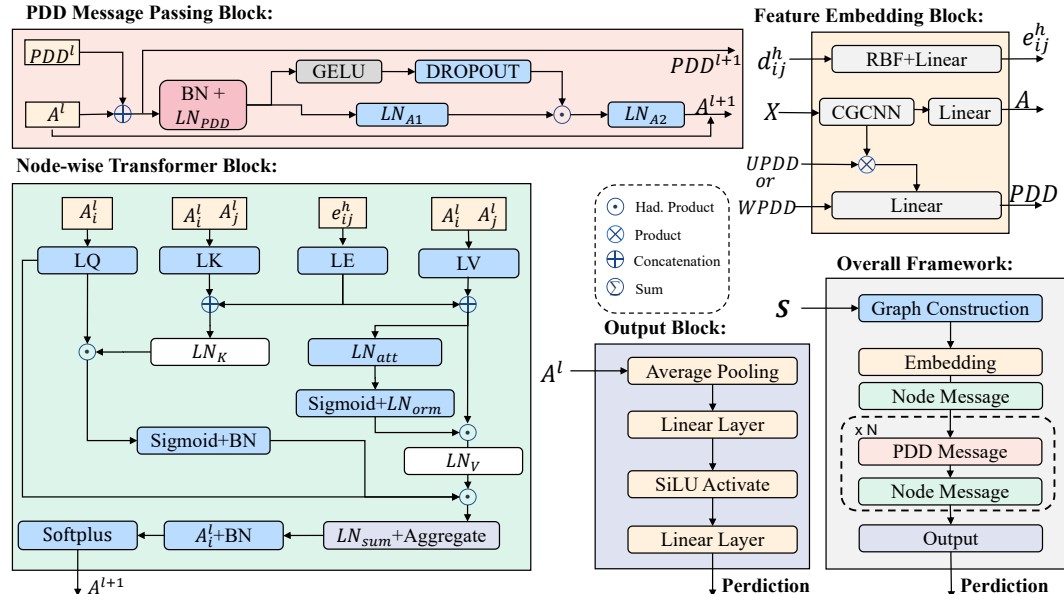

Figure 3: Architecture Overview. PDDFormer accepts an input crystal structure $S$. During the prediction process, it first undergoes a graph construction step to generate a continuous and complete crystal graph structure, followed by an embedding block, then multiple blocks of node-wise Transformer and PDD Message Passing, and finally, an output block.

**Node-wise transformer block.** Building upon the constructed graph, we aggregate the node information. Let $a_i^l$ represent the input feature vector of node $i$ at layer $l$ in PDDFormer. To better capture the importance of different atoms, we added an attention mechanism at the atomic level, $v_{ij}^l = v \odot sigmoid\left(LN_{orm}\left(LN_{att}\left(v\right)\right)\right)$. To enhance model convergence, we added residual connections to capture shallow-layer information, $m_{ij}^h = q_{ij}^l + sigmoid\left(BN\left(att^l\right)\right) \odot LN_V\left(v_{ij}^l\right)$. $LNorm$ denotes the layer normalization (Ba, 2016) operation. BN denotes the batch normalization layer (Ioffe, 2015). To validate the effectiveness of our modifications, we conducted ablation experiments in Appendix A.6.3.

Then, we obtain the message $M_i^l$ by aggregating the information from the neighborhood of node $i$ over multiple edges, and $A_i^{l+1}$ is realized as follows :

$$M_i^l = \sum_{j \in A_i} \sum_h LN_{sum}\left(m_{ij}^h\right), \; A_i^{l+1} = softplus\left(a_i^l + BN\left(M_i^l\right)\right) \tag{5}$$

where $LN_{msg}$ is the linear transformation used for updating the edge messages.

**PDD message passing block.** $\mathcal{A}^l$ and $PDD^l$ represent the atomic features and 3D periodic pattern encoding at layer $l$, respectively. Its message-passing mechanism is as follows:

$$\begin{aligned} \mathcal{PDD}^{l+1} &= \mathcal{PDD}^l + \mathcal{A}^{l+1}, \; A_1, A_2 = LN_{PDD}\left(BN\left(PDD^{l+1}\right)\right), \\ A^{l+1} &= A^l + LN_{A2}\left(LN_{A1}\left(A_1\right) \odot Drop\left(GELU\left(A_2\right)\right)\right) \end{aligned} \tag{6}$$

In this process, we update $A^{l+1}$ using residual connections (He et al., 2016).

Finally, we use average pooling to aggregate the features of all nodes in the graph, followed by a nonlinear layer, and then a linear layer to obtain the scalar output of the graph as described above. A detailed description of the PDDFormer architecture can be found in Appendix A.2.

## 5 EXPERIMENTS

We conducted experiments on two material benchmark datasets, namely the Materials Project (Chen et al., 2019) and Jarvis (Choudhary et al., 2020) datasets. Detailed descriptions of the datasets can be found in Appendix A.1. More information about the experimental settings of PDDFormer can be found in Appendix A.3. Baseline methods include CFID (Choudhary et al., 2018), CGCNN (Xie

Table 1: Comparison between UPDDFormer, WPDDFormer, and other baselines in terms of test MAE on the JARVIS dataset. The best results are shown in **bold** and the second best results are shown with underlines.

| Method | Formation Energy | Bandgap(OPT) | Total Energy | Ehull | Bandgap(MBJ) |
|--------|------------------|--------------|--------------|-------|--------------|
|        | eV/atom          | eV           | eV/atom      | eV    | eV           |
| CFID (2018)   | 14    | 0.30  | 240   | 220   | 0.53  |
| CGCNN         | 63    | 0.20  | 78    | 170   | 0.41  |
| SchNet        | 45    | 0.19  | 47    | 140   | 0.43  |
| MEGNET        | 47    | 0.145 | 58    | 84    | 0.34  |
| GATGNN        | 47    | 0.17  | 56    | 120   | 0.51  |
| ALIGNN        | 33.1  | 0.142 | 37    | 76    | 0.31  |
| M3GNet        | 39.0  | 0.145 | 41    | 95    | 0.36  |
| Matformer     | 32.5  | 0.137 | 35    | 64    | 0.30  |
| PotNet        | 29.4  | 0.127 | 32    | 55    | 0.27  |
| CrysMMNet     | 28.0  | 0.128 | 34    | –     | 0.278 |
| CrysDiff (2024) | 29.0 | 0.131 | 34   | 62    | 0.287 |
| Crystalformer | 30.6  | 0.128 | 32    | 46    | 0.274 |
| eComFormer    | 28.4  | 0.124 | 32    | 44    | 0.28  |
| iComFormer    | 27.2 | 0.122 | 28.8 | 47 | 0.26 |
| **UPDDFormer** | 27.6 | 0.127 | 29.4 | 35.6 | 0.269 |
| **WPDDFormer** | **26.9** | **0.120** | **28.2** | **32.6** | **0.251** |

& Grossman, 2018), SchNet (Schütt et al., 2017), MEGNET (Chen et al., 2019), GATGNN (Louis et al., 2020), ALIGNN (Choudhary & DeCost, 2021), M3GNet (Chen & Ong, 2022), Matformer (Yan et al., 2022), PotNet (Lin et al., 2023), CrysMMNet (Das et al., 2023), CrysDiff (Song et al., 2024), Crystalformer (Taniai et al., 2024), and ComFormer (Yan et al., 2024a). For all baselines on the material datasets, we report the results provided in the cited papers.

Table 2: Comparison of test MAE between UPDDFormer, WPDDFormer, and other baselines on the Materials Project dataset.

| Method | Formation Energy | Band Gap | Bulk Moduli | Shear Moduli |
|--------|------------------|----------|-------------|--------------|
|        | eV/atom          | eV       | log(GPa)    | log(GPa)     |
| CGCNN (2018)       | 31    | 0.292 | 0.047  | 0.077  |
| SchNet (2018)      | 33    | 0.345 | 0.066  | 0.099  |
| MEGNET (2019)      | 30    | 0.307 | 0.060  | 0.099  |
| GATGNN (2020)      | 33    | 0.280 | 0.045  | 0.075  |
| ALIGNN (2021)      | 22    | 0.218 | 0.051  | 0.078  |
| M3GNet (2022)      | 24    | 0.247 | 0.050  | 0.087  |
| Matformer (2022)   | 21.0  | 0.211 | 0.043  | 0.073  |
| PotNet (2023)      | 18.8  | 0.204 | 0.040  | 0.065  |
| CrysMMNet (2023)   | 20.0  | 0.197 | 0.038  | 0.062 |
| Crystalformer (2024) | 18.6 | 0.198 | 0.0377 | 0.0689 |
| eComFormer (2024)  | 18.16 | 0.202 | 0.0417 | 0.0729 |
| iComFormer (2024)  | 18.26 | 0.193 | 0.0380 | 0.0637 |
| **UPDDFormer** | 18.31 | 0.196 | 0.0393 | 0.0686 |
| **WPDDFormer** | **16.61** | **0.189** | **0.0336** | **0.0617** |

## 5.1 EXPERIMENTAL RESULTS

**JARVIS.** The quantitative results for JARVIS (Choudhary et al., 2020) are shown in Table 1. WPDDformer achieves the best performance across all tasks. Notably, WPDDFormer and UPDDFormer outperform eComFormer by 26% and 19% respectively in the Ehull task.

**The Materials Project (MP).** The experimental results on MP (Chen et al., 2019) are shown in Table 2. WPDDformer performs significantly better than previous works across all tasks, with a 10.8% improvement over the second-best model in the bulk moduli task. Additionally, the excellent prediction accuracy of WPDDFormer in the bulk modulus and shear modulus tasks, using only $4,664$ training samples, demonstrates the expressiveness and robustness of WPDD multi-edge crystal graphs under limited training samples.

Table 3: Efficiency comparison with ConFormer on the Jarvis Formation Energy task. We show the training time per epoch, total training time, time complexity, GPU memory consumption, and total number of parameters. The experiments were conducted using a 3090 RTX 24GB GPU.

| Models | Time/epoch | Total | GPU memory usage | Complexity | Model Para. |
|---|---|---|---|---|---|
| eConformer | 120s | 16.7h | 18GB | $O(nk)$ | 12.4M |
| iConformer | 129s | 25.0h | 12GB | $O(nk)$ | 5.0M |
| **WPDDFormer** | 98s | 10.9h | 8.5GB | $O(nk)$ | 6.76M |

Overall, our methods are compared with 14 existing methods across the two datasets. Our WPDDFormer consistently outperforms all methods in all tasks. Additionally, WPDDFormer shows a significant improvement in prediction accuracy compared to UP-PDFormer. This improvement is not only because the WPDD graph structure is complete and continuous, while UPPD can only ensure continuity, but also because UPPD requires dimensional alignment as mentioned in 4.6, which results in some loss of the expression of global information about the unit cell.

| Method | Num. Block | Ehull | Bulk |
|---|---|---|---|
| NO PDD Block | 4,0 | 39.2 | 0.0410 |
| without PDD | 4,3 | 36.3 | 0.0400 |
| **UPDDFormer** | 3,2 | 37.4 | 0.0446 |
| **UPDDFormer** | 4,3 | **35.6** | **0.0392** |
| **WPDDFormer** | 3,2 | 34.0 | **0.0336** |
| **WPDDFormer** | 4,3 | **32.6** | 0.0341 |

Table 4: Num. Block represents the number of Node-wise transformer blocks and PDD message passing blocks.

**Efficiency** This experiment reports the training and inference times for WPDDFormer and Con-Former using the best model configurations. We also report the total number of parameters for each model. As shown in Table 3, all these models have a time complexity of $O(nk)$, where $n$ represents the number of atoms in the unit cell and $k$ represents the average number of neighbors. The data in the table is averaged over three experiments. Although WPDDFormer has a higher parameter count compared to iConFormer, its training time overhead is significantly lower than that model, and it uses less GPU memory. Its memory usage is only $70.8\%$ of iConFormer and $47.2\%$ of eConFormer. This demonstrates that our WPDDFormer achieved significantly superior experimental results with lower computational cost and faster computation speed. Additional four tasks from the JARVIS dataset are documented in Appendix A.6.1.

### 5.2 Ablation studies

In this section, we demonstrate the impact of introducing (W/U)PDD on the representation learning of crystal materials through ablation studies. Specifically, we conducted experiments on the MP and JARVIS datasets, using testing mean absolute error (MAE) as the quantitative evaluation metric, comparing the results for **Bulk Moduli** and **Ehull** tasks, as shown in Table 4.

By comparing (W/U)PDDFormer models with different numbers of Node-wise Transformer Blocks and PDD Message Passing Blocks to models without (W/U)PDD information but retaining the $PDD$ message passing blocks, we validate the importance of (W/U)PDD. The results show that compared to models without the $PDD$ message passing blocks, WPDDForemer achieved improvements of $18.0\%$ and $16.8\%$ in the Bulk Moduli and Ehull tasks, respectively. Compared to models that retain only the $PDD$ message passing blocks but lack (W/U)PDD information, we achieved improvements of $16.0\%$ and $10.2\%$ in these two tasks, respectively.

### 6 Conclusion and future work

In this study, we integrated WPDD and UPDD into the representation of crystal structures, achieving a complete and continuous construction of crystal graphs. This resolves the ambiguity in crystal graph representations for predicting the properties of crystalline materials and bridges the gap between traditional crystal descriptors and dynamic atomic behavior. Experimental results demonstrate the significant advantage of our WPDDFormer in various property prediction tasks. Ensuring the completeness and continuity of crystal graphs after incorporating angular information is a problem that will be further explored in the future.

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

## A APPENDIX

### A.1 DATASET DESCRIPTIONS

In this section, we provide more detailed information about the JARVIS, and The Materials Project datasets.

**The Materials Project dataset.** Materials Project (MP) is a collection of $69,239$ materials from the Materials Project database retrieved byChen et al. (2019). We follow the experimental setup of Matformer (Yan et al., 2022) using the same training, validation, and test sets. For the formation energy and band gap tasks, the training, validation, and test sets contain crystals of $60,000$, $5,000$, and $4,239$, respectively. Among these, there are 38,344 samples with at least 20 atoms per unit cell, accounting for approximately $55.4\%$. There are $2,047$ samples with at least 100 atoms per unit cell, accounting for about $3.0\%$. We evaluate our Matformer on four key crystal property tasks: formation energy, band gap, bulk modulus, and shear modulus. For the bulk modulus and shear modulus tasks, the training, validation, and test sets contain $4,664$, $393$, and $393$ crystals, respectively.

**The JARVIS dataset.** JARVIS is a database proposed by Choudhary et al. (Choudhary et al., 2020). For the JARVIS dataset, we follow the approach of Matformer (Yan et al., 2022) and divide the data into training, validation, and test sets in an $8:1:1$ ratio. We evaluate our PDDFormer model on nine crucial crystal property tasks, including formation energy, bandgap (OPT), bandgap (MBJ), total energy, Ehull, bulk modulus (Kv), shear modulus (Gv), SLME (%), and Spillage. For the formation energy, total energy, and bandgap (OPT) tasks, the training, validation, and test sets contain $44,578$, $5,572$, and $5,572$ crystal samples, respectively. Among these, there are 8,089 samples with at least 20 atoms per unit cell, accounting for approximately $14.5\%$. Only 4 samples have at least 100 atoms per unit cell. For the Ehull task, these numbers are $44,296$, $5,537$, and $5,537$ samples; for the bandgap (MBJ) task, they are $14,537$, $1,817$, and $1,817$ samples; for bulk modulus (Kv) and shear modulus (Gv) tasks, they are $15,744$, $1,968$, and $1,968$ samples; for SLME (%) task, they are $7,254$, $906$, and $906$ samples; and for the Spillage task, they are $9,101$, $1,137$, and $1,137$ samples.

### A.2 PDDFORMER CONFIGURATIONS

We trained on the Formation Energy and Band Gap tasks of the MP dataset using an RTX A100 40GB GPU, and on the shear and bulk tasks of the JARVIS dataset and the MP dataset using an RTX 4090 24GB GPU.

**Notations.** $\mathcal{A} \in \mathcal{R}^{n \times da}$ is the atomic feature matrix obtained by embedding the atomic matrix $\mathcal{X} \in \mathcal{R}^{n \times 1}$ in the unit cell, where n represents the number of atoms in the unit cell, $\mathcal{A} = [a_1, a_2 \cdots a_{n-1}, a_n]^T \in \mathrm{R}^{n \times da}$, and $a_i$ represents the da-dimensional feature vector of atom $i$ in $\mathcal{A}$. $e_{ij}^h \in \mathcal{E}$ is the de-dimensional feature vector of the h-th edge connecting nodes $i$ and $j$. Typically, de is the same dimension as da. In constructing the PDD, we take the nearest neighbors $k = 92$, resulting in its dimensional information where PDD $\in \mathcal{R}^{n \times 92}$. The WPDD incorporates an additional dimension for atomic weights, thus its dimension is WPDD$\in \mathcal{R}^{n \times 93}$. For the UPDD, prior to alignment, its dimension is solely related to the number of atoms in the unit cell, expressed as UPDD$\in \mathcal{R}^{n \times n}$. After alignment, it matches the dimension of $\mathcal{A}$ to facilitate information aggregation. $\mathcal{XI} \in \mathcal{R}^{tn \times 2}$ is the index of the points corresponding to the edge, where $t$ is the number of the nearest edges aggregated within our cutoff radius (Flor et al., 2016).

**Graph embeddings.** For the two datasets we use, we employ the CGCNN atomic embedding, where the atomic number is mapped to a 92-dimensional embedding vector. Subsequently, we apply a linear transformation to map it to a 256-dimensional vector, serving as the input $a_i$ passed to the first PDDFormer message update block. For each edge, we employ 256 RBF kernels to map the Euclidean distance to a 256-dimensional embedding vector, with kernel centers ranging from $-4.0$ to $0.0$. It is then mapped to a 256-dimensional vector as the edge input $e_{ij}$, through a nonlinear layer followed by a linear layer. For the UPDD, after performing matrix multiplication with the embedded atomic information $\mathcal{A}$, it is passed through a nonlinear layer and a linear layer to map it to the same

dimension as $\mathcal{A}$. For the PDD and WPDD, they are directly processed through a nonlinear layer and a linear layer to map them to the same dimension as $\mathcal{A}$ and then passed to the message update layer.

**Settings of node-wise transformer block.** LQ, LK, LV, LE, $LN_{att}$, and $LN_{sum}$ are linear transformation layers that map 256-dimensional input features to 256-dimensional output features. $LN_{att}$ and $LN_{orm}$ are linear transformation layers that map $256 \times 3$ dimensional input features to $256 \times 3$ dimensional output features and layer normalization (Ba, 2016), respectively. $LN_K$ and $LN_V$ are nonlinear transformations for key and value, including one linear layer that maps the concatenated $256 * 3$ dimensional input features to 256-dimensional output features, one SiLU activation layer (Paul et al., 2022), and one linear layer that maps the 256-dimensional input features to 256-dimensional output features. Our Node-wise transformer module is inspired by the corresponding module in ConFormer (Yan et al., 2024a). The message from node $j$ to $i$ is formed by the corresponding query $q_{ij}^l$, key $k_{ij}^l$, and value features $v_{ij}^l$ as follows:

$$q_{ij}^l = LQ\left(a_i^l\right), \ k_{ij}^l = \left(LK\left(a_i^l\right) \oplus LK\left(a_j^l\right) \oplus LE\left(e_{ij}^h\right)\right),$$
$$v = \left(LV\left(a_i^l\right) \oplus LV\left(a_j^l\right) \oplus LE\left(e_{ij}^h\right)\right), \ att^l = \frac{q_{ij}^l \odot LN_K\left(k_{ij}^l\right)}{\sqrt{d_{q_{ij}^l}}}, \quad (7)$$

We use $\oplus$ and $\odot$ to denote concatenation and element-wise product. where LQ, LK, LV, and LE are the linear transformations for query, key, value, and edge features. $LN_K$, $LN_V$ are the nonlinear transformations for key and value, including two linear layers and an activation layer in between, and $d_{q_{ij}^l}$ is the dimension of $q_{ij}^l$

**Settings of PDD message passing block.** $LN_{PDD}$ is a linear transformation layer that maps 256 dimensional input features to 256-dimensional output features. Then, the first 128 dimensions are assigned to A1, and the remaining dimensions are assigned to A2. Dropout (Srivastava et al., 2014) is set to 0.1. After passing through $LN_{A1}$, which is a linear transformation layer that maps 128 dimensional input features to 128 dimensional output features, the data then goes through $LN_{A2}$, another linear transformation layer that maps 128 dimensional input features to 256 dimensional output features.

**Settings of the output block.** After the final layer of message passing, we aggregate the node features in the graph through mean pooling. Then, we use a linear layer to map the 256-dimensional graph-level features to 256-dimensional output features, followed by a SiLU activation layer. Then, we map the output to a scalar value through a linear transformation layer to complete our task.

### A.3 HYPERPARAMETER SETTINGS OF PDDFORMER

In this subsection, we present the detailed hyperparameter settings of WPDDFormer for different tasks. We slightly tuned the parameters of our method for the material datasets, and further adjustments are expected to yield higher performance in different tasks.

Table 5: Model settings of WPDDFormer for JARVIS dataset.

| Parameter | Learning rate | Num. neighbors | Epoch number | Num. Node and PDD |
|---|---|---|---|---|
| formation energy | 0.001 | 25 | 400 | 4,3 |
| band gap (OPT) | 0.0005 | 25 | 500 | 4,3 |
| band gap (MBJ) | 0.0005 | 18 | 300 | 4,3 |
| total energy | 0.001 | 25 | 500 | 4,3 |
| Ehull | 0.001 | 25 | 500 | 4,3 |
| Bulk Moduli(Kv) | 0.001 | 18 | 300 | 4,3 |
| Shear Moduli(Gv) | 0.001 | 18 | 300 | 4,3 |
| SLME(%) | 0.001 | 18 | 300 | 4,3 |
| Spillage | 0.0005 | 18 | 200 | 4,3 |

**JARVIS.** We show the model settings of WPDDFormer in Table 5. The evaluation metric for these tasks is the test mean absolute error (MAE), batch size of 64, weight decay (Loshchilov & Hutter,

2017) set to 1e-5. Specifically, the WPDDFormer was trained using the MAE loss function and the Adam optimizer (Kingma & Ba, 2014). For the formation energy, total energy, and Ehull tasks, the model was trained for 500 epochs with an initial learning rate set to 0.001; for the bulk modulus (Kv) and shear modulus (Gv) tasks, it was trained for 300 epochs with an initial learning rate set to 0.001. The band gap (MBJ) and Spillage tasks were trained for 300 and 200 epochs, respectively, with an initial learning rate set to 0.0005. For these eight tasks, the Onecycle scheduler (Smith & Topin, 2019) was used, with a pct start of 0.3. For the band gap (OPT) task, a polynomial scheduler was used for 500 epochs, with an initial learning rate of 0.0005 and a final learning rate of 0.00001. The parameter settings for UPDDFormer are the same as those for WPDDFormer across different tasks.

**The Materials Project.** We present the model settings for WPDDFormer in Table 6. For the Materials Project dataset, all models are trained using the MAE loss function, with a batch size of $64$ and weight decay (Loshchilov & Hutter, 2017) set to 1e-5. The Adam optimizer and Onecycle scheduler are used, with a pct start of 0.3. Specifically, the formation energy model is trained for 500 epochs with an initial learning rate of 0.001, the band gap model for 400 epochs with an initial learning rate of 0.001, and the bulk moduli and shear moduli models for 300 epochs each, with initial learning rates of 0.001 and 0.0001, respectively. The model settings for WPDDFormer in Table 7

Table 6: Model settings of WPDDFormer for The Materials Project dataset.

| Parameter | Learning rate | Num. neighbors | Epoch number | Num. Node and PDD |
|---|---|---|---|---|
| formation energy | 0.001 | 25 | 500 | 4,3 |
| band gap | 0.0005 | 25 | 500 | 4,3 |
| bulk moduli | 0.001 | 25 | 300 | 3,2 |
| shear moduli | 0.0001 | 25 | 300 | 4,3 |

Table 7: Model settings of UPDDFormer for The Materials Project dataset.

| Parameter | Learning rate | Num. neighbors | Epoch number | Num. Node and PDD |
|---|---|---|---|---|
| formation energy | 0.001 | 25 | 500 | 4,3 |
| band gap | 0.001 | 25 | 400 | 4,3 |
| bulk moduli | 0.001 | 16 | 300 | 4,3 |
| shear moduli | 0.001 | 16 | 300 | 4,3 |

### A.4 INVARIANCE PROPERTIES

#### A.4.1 DEFINITION

According to Yan et al. (2022), we represent a crystal structure with a triple (X, P, L), where $(X, P) \in U$, defined as follows: $X = [x_1, \ldots, x_N] \in \mathbb{R}^{d \times N}$ represents the states of N atoms in the unit cell, $P = [p_1, \ldots, p_N] \in \mathbb{R}^{3 \times N}$ denotes the 3D Cartesian coordinates of these atoms, $L = [\ell_1, \ell_2, \ell_3] \in \mathbb{R}^{3 \times 3}$ is the lattice vector matrix. The infinite crystal structure is:

$$\tilde{P} = \{\tilde{p}_i = p_i + h_1 l_1 + h_2 l_2 + h_3 l_3 \mid h_1, h_2, h_3 \in \mathbb{Z}, i \in \mathbb{Z}, 1 \leq i \leq n\},$$
$$\tilde{X} = \{\tilde{x}_i = x_i \mid i \in \mathbb{Z}, 1 \leq i \leq n\}$$

(8)

The coordinates of the n points are defined within the unit cell U as determined by L, meaning their fractional coordinates are $L^{-1}P \in [0, 1)^{3 \times N}$. When the overall network architecture is viewed as a function $f(X, P, L) \to \mathcal{X}$, they satisfy the following invariance properties.

The unique geometric prior knowledge of crystals includes two distinct physical constraints and symmetries: E(3) invariance within the unit cell and periodic invariance.

**Definition 7: Unit Cell E(3) Invariance.** Following Matformer Yan et al. (2022), A function $f : (\mathcal{X}, \mathcal{P}, \mathcal{L}) \to \mathcal{Y}'$ is unit cell $E(3)$ invariant if, for all $Q \in \mathbb{R}^{3 \times 3}$, where $|Q| = 1$, and $b \in \mathbb{R}^3$, we have $f(\mathcal{X}, \mathcal{P}, \mathcal{L}) = f(\mathcal{X}, Q\mathcal{P} + b, Q\mathcal{L})$.

Table 8: Comparison between PDD and WPDD in terms of test MAE on JARVIS dataset. The best results are shown in **bold**.

| | Formation Energy | Bandgap(OPT) | Total Energy | Ehull | Bandgap(MBJ) |
|---|---|---|---|---|---|
| Method | eV/atom | eV | eV/atom | eV | eV |
| PDDFormer | **26.5** | 0.124 | 28.7 | **32.3** | **0.244** |
| WPDDFormer | 26.9 | **0.120** | **28.2** | 32.6 | 0.251 |

In other words, the crystal structure remains unchanged when the position matrix $\mathcal{P}$ of the unit cell structure undergoes rotation, translation, or reflection.

Moreover, different minimal repeatable structures can be used to represent the same crystal. These different crystal structure representations $(X, P, L)$ introduce a constraint known as periodic invariance. Two periodic transformations can generate different minimal unit cell representations for the same crystal structure, including shifting the periodic boundary and changing the periodic pattern while maintaining the same unit cell volume.

**Definition 8: Periodic Invariance.** Following Matformer Yan et al. (2022), A function $f : (\mathcal{X}, \mathcal{P}, \mathcal{L}) \to \mathcal{Y}'$ is periodically invariant if, for any possible minimal unit cell representation $\mathcal{M}' = (\mathcal{X}', \mathcal{P}', \mathcal{L}')$ of a given infinite crystal structure $(\bar{\mathcal{X}}, \bar{\mathcal{P}})$, we have $f(\mathcal{X}, \mathcal{P}, \mathcal{L}) = f(\mathcal{X}', \mathcal{P}', \mathcal{L}')$.

### A.4.2 PROOFS OF INVARIANCE

**Proof of Unit Cell E(3) Invariance and Periodic Invariance.** If the PDD multi-edge graph we construct exhibits E(3) invariance and periodic invariance, then every step in the crystal graph construction process must conform to the crystal constraints. Therefore, we analyze the construction process of the crystal graph to progressively demonstrate E(3) invariance and periodic invariance.

First, we construct a crystal graph with $n$ nodes using a minimal unit cell structure containing $n$ atoms. This step has been handled by the JARVIS and MP datasets. Since all minimal unit cell structures for a given crystal share the same number of atoms and corresponding atomic features, this step is E(3) invariant and periodically invariant.

After determining the selection of atoms, we begin to establish edge information connecting neighboring nodes for each atom. An edge is established from node $j$ to node $i$ when the Euclidean distance $|e_{j'i}|^2$ between a duplicate of $j$ and $i$ satisfies $|e_{j'i}|^2 = |p_j + k_1' l_1 + k_2' l_2 + k_3' l_3 - p_i|^2 \leq r$, where $r \in \mathbb{R}$ is the cutoff radius. We select the nearest $t$ edges within the cutoff radius, each with a corresponding edge feature $|e_{j'i}|^2$. The Euclidean distance $|e_{j'i}|^2$ between duplicates $j$ and $i$ remains invariant under E(3) transformations and different representations of the unit cell structure. Thus, the neighborhood information for node $i$ is E(3) invariant and periodically invariant.

Finally, we establish the crystal structure representations for WPDD and UPDD. For WPDD, we select $k$ nearest neighbors based on Euclidean distance to create the corresponding WPDD row for each node $i$. For UPDD, we center around node $i$ and select atoms from the reconstructed unit cell to create the corresponding UPDD row for each node $i$ based on Euclidean distance. The Euclidean distance remains invariant under E(3) transformations and different unit cell structures. Thus, the PDD row of node $i$ is both E(3) invariant and periodic invariant.

By combining these three steps in the construction process of crystal graphs, we complete the proof that the proposed PDD crystal graph representation is E(3) invariant and periodic invariant.

### A.5 PDD AND WPDD

In this chapter, we investigate the impact of atomic weight distribution (W) in $WPDD = (W, PDD)$ on the experimental results of crystal property prediction. We conduct experiments on the JARVIS and MP datasets, comparing the effects of WPDD with those of PDD without atomic weight distribution.

By comparing the data in Tables 8 and 9, it is evident that using WPDD or PDD for experiments yields mixed results, indicating that the atomic weight distribution (W) in WPDD does not have a

Table 9: Comparison of test MAE between PDD and WPDD on the Materials Project dataset.

| Method | Formation Energy eV/atom | Band Gap eV | Bulk Moduli log(GPa) | Shear Moduli log(GPa) |
|---|---|---|---|---|
| PDDFormer | 16.96 | **0.187** | 0.0337 | 0.0628 |
| WPDDFormer | **16.61** | 0.189 | **0.0336** | **0.0617** |

significant impact on the experimental outcomes. The reason is that when the model transmits information through the PDD message passing module, we sum the atomic information A with PDD and assign it to PDD. This means that PDD implicitly contains atomic information, so the introduction of atomic weight distribution as atomic information has a minimal effect on the experimental results.

## A.6 MORE EXPERIMENTAL

### A.6.1 JARVIS.

As shown in Table 10, WPDDFormer also outperforms all other baseline models in these four tasks, achieving the best results in three out of four tasks and second-best in one. For the Bulk Moduli (Kv) and Shear Moduli (Gv) tasks, $19,680$ training samples were used, and for the SLME (%) and Spillage tasks, $9,066$ and $11,375$ training samples were used, respectively. WPDDFormer demonstrates its adaptability to tasks with varying data scales.

| **Method** | Bulk Moduli(Kv) GPa | Shear Moduli(Gv) GPa | SLME(%) No unit | Spillage No unit |
|---|---|---|---|---|
| CGCNN | 14.47 | 11.75 | 8.022 | 0.454 |
| SchNet | 14.33 | 10.67 | – | – |
| MEGNET | 15.11 | 13.09 | – | – |
| GATGNN | 14.32 | 12.48 | 7.504 | 0.431 |
| ALIGNN | 10.40 | 9.481 | 5.145 | 0.389 |
| Matformer | 11.21 | 10.76 | 5.260 | 0.398 |
| CrysMMNet | 9.625 | **8.471** | – | – |
| PotNet | 10.06 | 8.883 | – | – |
| CrysDiff | 9.875 | 9.193 | 5.030 | **0.358** |
| eComFormer | 10.79 | 9.826 | 4.610 | 0.373 |
| iComFormer | 9.617 | 9.098 | 4.583 | 0.360 |
| **UPDDFormer** | 10.13 | 9.143 | 4.566 | 0.377 |
| **WPDDFormer** | **9.546** | 8.808 | **4.300** | **0.358** |

Table 10: Comparison between WPDDFormer, UPDDFormer, and other baselines in terms of test MAE on the JARVIS dataset. The best results are shown in **bold** and the second best results are shown with underlines. The results reported for PotNet and Conformer in the table are those obtained from training using their published code.

### A.6.2 THE NUMBER OF NEIGHBORS ( K ) OF WPDD

In this section, we investigate the effect of the number of neighbors with different cutoff radii on the WPDD experiment. In the main text, we set the number of neighbors $k = 92$, which matches the dimensionality of the atomic feature embeddings used by CGCNN. However, in practical applications, it is necessary to determine a sufficiently large $k$ in advance to ensure completeness for any test crystal, especially in extreme cases where $k$ must be greater than the number of atoms in any test crystal (Yan et al., 2024a). Therefore, to ensure completeness on the JARVIS and MP datasets, we calculated the maximum number of atoms in each dataset, which is $140$ for JARVIS, $152$ for MP's bulk and shear, and $296$ for the rest. As a result, in the experiments, the maximum number of neighbors for the JARVIS and MP datasets with different cutoff radii were selected as $150$, $160$, and $300$, respectively.

Table 11: Comparison of test set MAE for WPDD with different numbers of neighbors $k$ on the JARVIS dataset. The best results are shown in **bold**.

| | | Formation Energy | Bandgap(OPT) | Total Energy | Ehull | Bandgap(MBJ) |
|---|---|---|---|---|---|---|
| Method | k | eV/atom | eV | eV/atom | eV | eV |
| WPDDFormer | 60 | **26.5** | 0.122 | **27.9** | 33.3 | 0.266 |
| **WPDDFormer** | 92 | 26.9 | **0.120** | 28.2 | **32.6** | **0.251** |
| WPDDFormer | 150 | 27.2 | 0.121 | 29.0 | 34.0 | 0.254 |

Table 12: Comparison of test set MAE for WPDD with different numbers of neighbors $k$ on the Materials Project dataset.

| | | Formation Energy | Band Gap | | Bulk Moduli | Shear Moduli |
|---|---|---|---|---|---|---|
| Method | k | eV/atom | eV | k | log(GPa) | log(GPa) |
| **WPDDFormer** | 92 | **16.61** | 0.189 | 92 | 0.0336 | **0.0617** |
| WPDDFormer | – | – | – | 120 | **0.0295** | 0.0647 |
| WPDDFormer | 300 | 16.83 | **0.186** | 160 | 0.0303 | 0.0652 |

From the experimental results in Tables 11 and 12, it can be seen that the choice of neighbors with different cutoff radii causes fluctuations in the model's performance. However, its performance still shows the best results compared to the other models presented in the main text.

**efficency**  We report the training time per epoch, total training time, inference time, time complexity, GPU memory consumption, and total number of parameters for WPDDFormer and UP-DDFormer using the best model configurations, comparing their efficiency on the JARVIS formation energy task. As shown in Table 13, all these models have a time complexity of $O(nk)$, where $n$ represents the number of atoms in the unit cell and $k$ represents the average number of neighbors. The data in the table is averaged over three experiments. Since WPDD and UPDD use the same model for experiments, with only the crystal graph construction differing, they have the same time complexity, nearly identical GPU memory consumption, total parameters, training time per epoch, and total training time. However, there is a significant difference in inference speed, with UPDD being **3.4** times faster than WPDD. The reason for this is in the data preprocessing stage, where WPDD requires more neighbor information, resulting in longer extraction times and slower inference speed. As the number of neighbors chosen with the increasing cutoff radius grows, the time for data preprocessing also increases, leading to a slowdown in inference speed.

Table 13: Efficiency comparison between UPDDFormer and WPDDFormer with different numbers of neighbors on the JARVIS formation energy task. The experiments were conducted using a 3090 RTX 24GB GPU.

| Models | k | Time/epoch | Total | GPU memory | Complexity | Model Para. | **inference** |
|---|---|---|---|---|---|---|---|
| WPDDFormer | 60 | 97s | 10.8h | 8.5GB | $O(nk)$ | 6.75M | 1014.2s |
| WPDDFormer | 92 | 98s | 10.9h | 8.5GB | $O(nk)$ | 6.76M | 1191.7s |
| WPDDFormer | 150 | 100s | 11.1h | 8.5GB | $O(nk)$ | 6.78M | 1504.7s |
| **UPDDFormer** | – | 95s | 10.6h | 8.5GB | $O(nk)$ | 6.76M | **351.3s** |

### A.6.3  MORE ABLATION

To verify the effectiveness of incorporating PDD descriptors into crystal graph construction, we present in Table 14 the impact on experimental results on the JARVIS dataset. Without changing the parameters of the WPDDFormer model, we investigated the effect of removing WPDD information on the experiments.

Table 14: Whether to include a comparison of WPDD's test MAE on the JARVIS dataset. The best results are shown in **bold**.

|  | Formation Energy | Bandgap(OPT) | Total Energy | Ehull | Bandgap(MBJ) |
|---|---|---|---|---|---|
| Method | eV/atom | eV | eV/atom | eV | eV |
| No PDD | 28.3 | 0.121 | 29.8 | 35.6 | 0.265 |
| WPDDFormer | **26.9** | **0.120** | **28.2** | **32.6** | **0.251** |

Table 15: The experimental results obtained on the JARVIS dataset. The best results are shown in **bold**.

|  | Formation Energy | Bandgap(OPT) | Total Energy | Ehull | Bandgap(MBJ) |
|---|---|---|---|---|---|
| Method | eV/atom | eV | eV/atom | eV | eV |
| Matformer | 32.5 | 0.137 | 35 | 64 | 0.30 |
| Matformer+WPDD | **31.1** | **0.131** | **32** | **56** | **0.27** |

Additionally, we applied PDD to other models. Table 15 shows the results of incorporating the pairwise distance distribution (PDD) into the Matformer architecture.

Overall, the results in Figures 14 and 15 effectively demonstrate the generalization capability of PDD, indicating that incorporating it into different model architectures can significantly improve the prediction accuracy of the original models.

In our model, the Node-wise Transformer Block is inspired by the Node-wise Transformer module of ConFormer. We made improvements to this design, and to demonstrate the generalization ability of our enhanced Transformer module, we conducted the following two experiments.

Experiment 1: We replaced our model's Node-wise Transformer Block (PT) with the node-wise transformer module from ConFormer (CT). As shown in the table 16, the WPDD model using our improved transformer module exhibited significant performance improvement

Experiment 2: We applied the improvements made to the transformer to ConFormer and conducted experiments on the JARVIS dataset. The results, as shown in the figure 17, indicate that the performance achieved significant improvements across all property prediction tasks.

Overall, the results of Experiments 1 and 2 demonstrate that the modifications we made are simple yet highly effective.

## A.7 CONTINUOUS TOLERANCE $\mathcal{T}$

Given that the experimentally measured unit cell and atomic coordinates are inevitably affected by atomic vibrations and measurement noise, slight perturbations in the atomic coordinates may occur. Therefore, during the construction of the multi-edge crystal graph, when selecting the t nearest edges within a cutoff radius, the chosen neighboring nodes may change, as shown in Figure 4 (a) and (b). This variation in neighboring nodes $j$ leads to changes in the atomic information of the neighboring nodes, resulting in discontinuities in the construction of the multi-edge crystal graph. To eliminate the influence of atomic perturbations on the neighbor selection and ensure the continuity of the

Table 16: The experimental results obtained on the JARVIS dataset. The best results are shown in **bold**.

|  | Formation Energy | Bandgap(OPT) | Total Energy | Ehull | Bandgap(MBJ) |
|---|---|---|---|---|---|
| Method | eV/atom | eV | eV/atom | eV | eV |
| WPDDFormer+CT | 28.1 | 0.122 | 30.4 | 34.7 | 0.253 |
| WPDDFormer | **26.9** | **0.120** | **28.2** | **32.6** | **0.251** |

Table 17: The experimental results obtained on the JARVIS dataset. The best results are shown in **bold**.

| Method | Formation Energy eV/atom | Bandgap(OPT) eV | Total Energy eV/atom | Ehull eV | Bandgap(MBJ) eV |
|---|---|---|---|---|---|
| iConformer | 28.1 | 0.122 | 30.4 | 34.7 | 0.253 |
| iConFormer+PT | **26.5** | **0.120** | **28.0** | **38** | **0.258** |

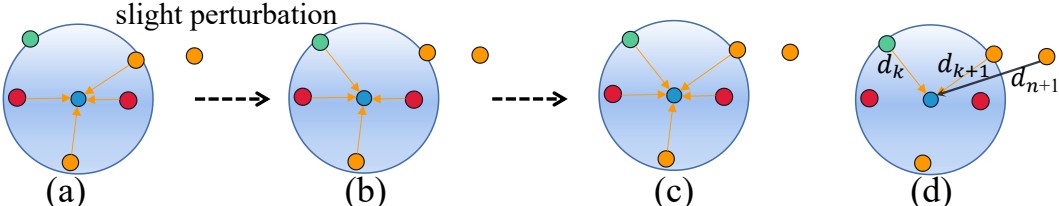

Figure 4: The different neighbor selection under slight perturbations.

crystal graph under perturbations, we define the concept of continuous tolerance $\mathcal{T}$ to guarantee the continuity of the constructed crystal graph.

Since the distances of atomic perturbations are typically on the order of sub-angstrom (Å), specifically, in common atomic structures or crystals, slight perturbations are generally less than $\mathcal{T} < 10^{-2}$. For larger perturbations (e.g., $> \mathcal{T} = 10^{-2}$), the continuity issue may no longer be effective. Therefore, when selecting neighbors, we can set a continuous tolerance value $\mathcal{T}$ in advance. When we select the t nearest edges within the cutoff radius if the distance of the (t+1)-th edge from node $i$ minus the distance of the t-th edge is less than the continuous tolerance, i.e., $d_{k+1} - d_k < T$, we include this neighbor in the graph construction as well. This process continues until the distance of the (n+1)-th edge from node i minus the distance of the n-th edge exceeds the continuous tolerance cutoff, i.e., $d_{n+1} - d_n > \mathcal{T}$. The n neighbors at this point are the selected nodes for constructing the crystal graph, as shown in Figure 4 (d), resulting in the final neighbor selection shown in Figure 4 (c). This approach ensures that the neighbor selection in the crystal graph construction does not change under atomic perturbations, and by using Lemma 1, we prove the continuity of the (W/U)PDD crystal graph we have constructed.

## A.8 PROOF OF INTEGRITY FOR WPDD CRYSTAL GRAPHS.

In this section, we provide a simple demonstration to showcase the integrity of our WPDD crystal graphs for any crystalline material. As shown in Figure 5, (a), (b), and (c) represent unstable crystal structures with identical crystal configurations but different atomic types. We present a straightforward example by setting the number of neighbors in PDD to $k = 4$ for the demonstration. We assume that the black atoms are carbon (Si) and the green atoms are oxygen (O).

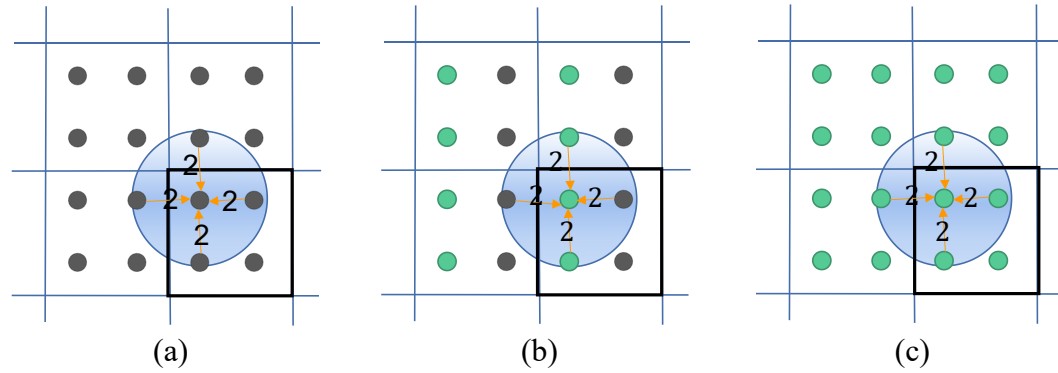

Figure 5: Unstable crystal structures with different element types but the same crystal structure.

Crystal structures can be classified into stable and unstable types and further divided into the following three categories: 1. Stable Crystal Structures(i.e., where no two crystals can have the same structure with only a difference in atomic types). 2. Unstable Crystal Structures with Differences in Atomic Coordinates. 3. Unstable Crystal Structures with Identical Structures but Differences in Atomic Types. We define completeness as follows: if the constructed crystal graph representation can differentiate between any two crystal structures that are not the same, we consider the representation to be complete. We will demonstrate the integrity of these three types of crystals to prove that our constructed WPDD crystal graph is complete.

The integrity of stable crystal structures and unstable crystal structures with differences in atomic coordinates can be ensured by the PDD (Periodic Crystal Descriptor), which is specifically designed to distinguish between different crystal structures. However, for unstable crystal structures with identical crystal structures (i.e., completely identical coordinates) but differences in atomic types, it cannot distinguish them(That is, the third category of crystal structures.). Therefore, PDD is not absolutely complete. This is because PDD was originally designed to describe stable crystal structures and cannot distinguish crystal structures with identical atomic coordinates but different atomic types (i.e., unstable crystal structures). As shown in Figure 5, it produces identical PDD matrices, PDD = (1, 2, 2, 2, 2), for the structures depicted in (a), (b), and (c), which have different atomic compositions. This limitation arises because the construction of PDD does not consider atomic types.

To address this issue, we construct WPDD multi-edge graphs that capture differences in atomic types, resolving this limitation. Details are provided below.

We address the representation of atomic types from the following two aspects, effectively resolving the aforementioned issue. First, we improved PDD by incorporating atomic information weights $\mathcal{W} = [w_1, \ldots, w_n]^T$, where $w_i = \frac{t(x_i)}{\sum_{j=1}^{n} t(x_j)}$. This allows us to construct WPDD = (W, PDD).

In addition, we do not use WPDD alone for predicting the properties of crystalline materials. Instead, we incorporate it as global information into the construction of a multi-edge crystal graph to better encode atomic information, such that the WPDD crystal graph $\mathcal{G}$ is represented as $\mathcal{G} = (\mathcal{X}, \mathcal{XI}, \mathcal{E})+$ WPDD, where $\mathcal{X}$ represents atomic information embedded through CGCNN, $\mathcal{E}$ represents edge information, and $\mathcal{XI}$ represents the information of the starting and ending nodes of the edges. This indicates that for any atom in the unit cell, we need to construct a WPDD row vector and perform the corresponding atomic information embedding. This ensures that for any two crystal structures with identical crystal structures but differing atomic types at corresponding coordinates, the $(\mathcal{X}, \mathcal{XI}, \mathcal{E})$ in their WPDD crystal graphs will differ. On the contrary, if two crystals have the same WPDD crystal graph representation, it means they share the same WPDD and multigraph representations. This indicates that their crystal structures and the atomic information at corresponding coordinates are identical, thus confirming that they are the same crystal. This contradicts our premise. For example, as shown in Figures 5 (a), (b), and (c).

Therefore, we can conclude that our WPDD crystal graph can identify all crystal structures.

