# OpenReview forum: "PDDFormer: Pairwise Distance Distribution Graph Transformer for Crystal Material Property Prediction"
_ICLR.cc/2025/Conference — ICLR 2025 Conference Withdrawn Submission_

### Official Review · Reviewer_xGYr · 2024-10-18

**Soundness:** 3
**Presentation:** 4
**Contribution:** 3
**Rating:** 6
**Confidence:** 3

**Summary:**

This paper presents atom-Weighted Pairwise Distance Distribution (WPDD) and Unit cell Pairwise Distance Distribution (UPDD) to characterize periodic crystal structure while accounting for atomic information. The continuity and geometrical completeness of these PDDs are theoretically evaluated. It then incorporates them into crystal graphs and develops graph transformer architectures for crystal property prediction. Comparative experiments show their better accuracy and efficiency over previous ML models, and ablation studies validate the importance of (W/U)PDD.

**Strengths:**

- The proposed (W/U)PDDs incorporate atomic information into PDD and address its high computational cost.
- The paper is overall well-written with nice flow, clarity, and illustrations.

**Weaknesses:**

- The experimentation focuses on relatively simple scalar properties, some of which do not physically depend on crystal periodicity.
- Incorporating PDD into crystal graphs makes the data physics-informed, however, the proposed model does not seem to consider interpretability.
- Minor issues
  - In Definition 1, I suggest using boldface or other methods to help distinguish scalar, vector, and matrix.
  - Some languages in Definitions are vague: (1) Line 133, the range of what “crystal graph” refers to in this context should be specified. (2) Line 137, the description of geometrical completeness is unclear.
  - Typos, e.g., Line 129 “and If” and Line 512, “However”.

**Questions:**

- UPDD has multiple drawbacks compared to WPDD and shows inferior performance in experiments. What’s its advantage and when is it preferred over WPDD?
- The source of improvement by incorporating PDD is unclear. Is there a way to investigate whether it is because of accounting for periodicity, or because of input being more informative in other ways?
- Would the authors consider evaluating the interpretability of (W/U)PDD crystal graphs and/or the proposed transformer models?

---

> ### Author Response · Authors · 2024-11-17
>
> Thank you for your valuable time and for recognizing the overall clarity of our paper. We will respond to your concerns and hope to address them satisfactorily. We look forward to your reply.
>
> >**Weakness 1:** The experimentation focuses on relatively simple scalar properties, some of which do not physically depend on crystal periodicity.
>
> **Answer 1:**  We agree with your observation that some simple properties in the experiments, such as the Ehull property in the JARVIS dataset, do not rely on the periodicity of crystals. However, most properties, such as Formation Energy, Bandgap (OPT), Bandgap (MBJ), and Total Energy, are to some extent dependent on the periodicity of the crystal. Therefore, representing periodic structures remains highly significant.
>
> >**Weakness 2:** Incorporating PDD into crystal graphs makes the data physics-informed, however, the proposed model does not seem to consider interpretability.
>
> **Answer 2:**  Although we did not explicitly state the information implied by PDD, its design purpose is to distinguish different crystal structures. PDD has successfully achieved effective differentiation of all crystal structures in the world’s largest materials collection—the Cambridge Structural Database—indicating its ability to capture the periodic characteristics of different crystals.
> Meanwhile, **(Gong et al.)[1]** suggested that hand-crafted descriptors are based on human understanding of material composition and structure and that GNNs may be sufficient to capture periodic features when the periodic length of the crystal (i.e., the length of the lattice vectors) is smaller than the perceptual domain of the atoms. However, the construction scope of a descriptor like PDD extends beyond the limitations of a unit cell. Therefore, we can infer that PDD can capture the periodic features of crystals to a certain structure.
>
> 3:** Minor issues**
>
> **3.1** In Definition 1, I suggest using boldface or other methods to help distinguish scalar, vector, and matrix.
>
> We have adopted the reviewer’s suggestions and revised the content of the paper to make it clearer and more comprehensible. Matrices, such as $\mathcal{PDD}$, $\mathcal{U}$, and $\mathcal{L}$, are represented in italics; vectors, such as **pi**, are represented in bold; and scalars, such as k and $d_{i1}​$, are represented in regular font.
>
> **3.2** Some languages in Definitions are vague: (1) Line 133, the range of what “crystal graph” refers to in this context should be specified. (2) Line 137, the description of geometrical completeness is unclear.
>
> (1) In line 133, the term "crystal graph" refers to the graph representation constructed to characterize crystal structures. We have revised these two definitions to make them clearer. The updated version of the paper has been uploaded, with the relevant changes highlighted in blue font.
>
> **3.3** Typos, e.g., Line 129 “and If” and Line 512, “However”.
>
> We appreciate the reviewer’s careful examination. We have corrected these two errors in the main text.
>
> >**Question 1:** UPDD has multiple drawbacks compared to WPDD and shows inferior performance in experiments. What’s its advantage and when is it preferred over WPDD?
>
> **Answer 1:**  Compared to WPDD, UPDD demonstrates relatively weaker performance. However, it still achieves competitive results compared to other models. The table below shows that UPDD has a significant advantage over WPDD in inference speed, **the inference speed of UPDD is 3.4 times faster than WPDD**. Therefore, in scenarios with high inference speed, UPDD can be used as a quicker alternative to facilitate experiments.
>
>
> A comparison of inference times for WPDDFormer and UPDDFormer with different numbers of neighbors k on the formation energy task in the JARVIS dataset.
>
> | Method | k | inference time|
>  |------|------|------|
> |WPDDFormer |60 |1014.2s|
> |WPDDFormer|92| 1191.7s |
> |WPDDFormer |150 | 1504.7s |
> |**UPDDFormer**|...|**351.3s**|
>
> This difference arises from the data preprocessing stage, where WPDD requires more neighbor information, resulting in longer extraction times and slower inference speed. As the cutoff radius increases and the number of selected neighbors grows, the data preprocessing time increases, leading to a further decrease in inference speed.
>
> >**Question 2:** The source of improvement by incorporating PDD is unclear. Is there a way to investigate whether it is because of accounting for periodicity, or because of input being more informative in other ways?
>
> **Answer 2:**  Please refer to our responses to **Weakness 2** and **Question 3** regarding the interpretability of PDD.
>
> **reference**
>
> [1] Gong S, Yan K, Xie T, et al. Examining graph neural networks for crystal structures: limitations and opportunities for capturing periodicity[J]. Science Advances, 2023, 9(45): eadi3245.

---

> ### Author Response · Authors · 2024-11-17
>
> >**Question 3:** Would the authors consider evaluating the interpretability of the (W/U)PDD crystal graph?
>
> **Answer 3:** To the best of our knowledge, there is currently a lack of effective methods to evaluate the interpretability of complex descriptors such as PDD (i.e., W/U-PDD crystal graphs). However, such complex descriptors may partially encapsulate information from lower-order descriptors, such as the lattice parameters (a, b, c) and their angles (α, β, γ).
>
> In the future, we plan to further investigate this issue through the following steps:
> - **Introduce simple descriptors:** Conduct experiments to observe their impact on crystal material property predictions.
> - **Analyze weights of these descriptors:** Determine which lower-order descriptors play a significant role in predicting crystal properties.
> - **Incorporate complex descriptors like PDD:** Introduce these into corresponding prediction tasks and observe whether they achieve better results. This will help assess whether complex descriptors inherently contain information from lower-order descriptors.
>
> Through this series of studies, we aim to enhance the interpretability of complex descriptors such as PDD, thereby providing deeper theoretical support for crystal material property predictions.
>
> >**Question 4:** Would the authors consider evaluating the interpretability of the proposed transformer models?
>
> **Answer 4:** For the interpretability issue of our proposed improved transformer model, which pertains to the interpretability of transformer models, this problem has been widely validated. We surveyed the existing literature on transformer interpretability.
>
> **(Elhage et al.)[1]** analyzed the operational principles of various modules in transformers through reverse engineering, progressing from simple to complex, and attempted to summarize and generalize them from a mathematical perspective.
>
> **(Ma et al.)[2]** proposed a novel interpretable visualization method to analyze and interpret the key attention interactions between Transformer patches.
>
> **(Fantozzi et al.)[3]** categorized the existing literature on Transformer interpretability and conducted a comprehensive analysis of Transformer interpretability from four perspectives: Activation, Attention, Gradient, and Perturbation.
>
> **Therefore, reviewers need not be concerned about the interpretability of the transformer architecture.**
>
> In summary, interpretability in the field of deep learning is a systematic and challenging issue. However, this paper focuses on addressing application-specific problems related to the completeness and continuity of crystal structures. Therefore, for this work, we do not currently consider the evaluation of interpretability. In the future, we will conduct in-depth research on this issue to provide more robust theoretical support for predicting the properties of crystalline materials.
>
> **reference**
>
> [1] Elhage N, Nanda N, Olsson C, et al. A mathematical framework for transformer circuits[J]. Transformer Circuits Thread, 2021, 1(1): 12.
>
> [2] Ma J, Bai Y, Zhong B, et al. Visualizing and understanding patch interactions in vision transformer[J]. IEEE Transactions on Neural Networks and Learning Systems, 2023.
>
> [3] Fantozzi P, Naldi M. The Explainability of Transformers: Current Status and Directions[J]. Computers, 2024, 13(4): 92.

---

> ### Comment · Reviewer_xGYr · 2024-11-19
>
> I appreciate the time and effort the authors take to improve the paper. Most of my concerns have been well addressed. There are two remaining issues:
> - The interpretability-related questions (W2, Q2-4) are discussed in the Rebuttal but not substantially addressed, which is fine at this point, but I cannot raise my rating because of that.
> - In "Minor Issues" 1, I meant to suggest clearly and formally defining "crystal graph". My understanding is that here it refers to graphs with nodes representing atoms and edges representing nearest neighbors; there are also other ways of "graph representation constructed to characterize crystal structures", e.g., [atomistic line graph](https://doi.org/10.1038/s41524-021-00650-1). The range where the continuity and completeness analyses apply should be clarified.

---

> ### Author Response · Authors · 2024-11-20
>
> Thank you for your valuable time and thorough review.
>
> >**Question 1:** The interpretability-related questions (W2, Q2-4) are discussed in the Rebuttal but not substantially addressed, which is fine at this point, but I cannot raise my rating because of that.
>
> We understand your concerns. But, the issue of interpretability is not the focus of this work. In future studies, we will conduct an in-depth investigation into this issue to provide stronger theoretical support for predicting the properties of crystalline materials.
>
> >**Question 2:**  My understanding is that here it refers to graphs with nodes representing atoms and edges representing nearest neighbors; there are also other ways of "graph representation constructed to characterize crystal structures", e.g., atomistic line graph. The range where the continuity and completeness analyses apply should be clarified.
>
> **The crystal graph referred to in the definition is a broad concept, encompassing all graph representations designed to characterize crystal structures.** This includes the multi-edge crystal graph constructed by CGCNN[1], the atomic line graph of ALIGNN[2] that incorporates bond angles, the crystal graphs with periodic encoding in Matformer[3], Crystalformer[4] and Conformer[5], and the PDD multi-edge crystal graph proposed in this paper. As long as the constructed crystal graph satisfies the definitions of continuous and complete graphs, it can be recognized as a complete and continuous crystal graph representation.
>
> We have reorganized the proof section on the completeness and continuity of crystal graphs in the paper, with the relevant text highlighted in blue.
>
> **reference**
>
> [1] Xie T, Grossman J C. Crystal graph convolutional neural networks for an accurate and interpretable prediction of material properties[J]. Physical review letters, 2018, 120(14): 145301.
>
> [2] Choudhary K, DeCost B. Atomistic line graph neural network for improved materials property predictions[J]. npj Computational Materials, 2021, 7(1): 185.
>
> [3] Yan K, Liu Y, Lin Y, et al. Periodic graph transformers for crystal material property prediction[J]. Advances in Neural Information Processing Systems, 2022, 35: 15066-15080.
>
> [4] Taniai T, Igarashi R, Suzuki Y, et al. Crystalformer: infinitely connected attention for periodic structure encoding[J]. arXiv preprint arXiv:2403.11686, 2024.
>
> [5] Yan K, Fu C, Qian X, et al. Complete and efficient graph transformers for crystal material property prediction[J]. arXiv preprint arXiv:2403.11857, 2024.

---

> > ### Comment · Reviewer_xGYr · 2024-11-21
> >
> > Thanks for the clarification regarding my Question 2. I've raised the Presentation score to 4.

---

> > > ### Author Response · Authors · 2024-11-24
> > >
> > > Dear reviewer  xGYr,
> > >
> > > We sincerely appreciate the time and effort you have dedicated to reviewing our work, as well as your valuable feedback and constructive suggestions.
> > >
> > > Yours sincerely, authors

---

### Official Review · Reviewer_uw9o · 2024-10-28

**Soundness:** 2
**Presentation:** 2
**Contribution:** 2
**Rating:** 3
**Confidence:** 4

**Summary:**

This paper introduces PDDFormer, a model designed to construct geometrically complete and invariant representations of crystals for the task of crystal property prediction. However, the completeness claim of the proposed representation is problematic, as it relies on the assumption that "the PDD is a generally complete invariant" (citing Widdowson & Kurlin, 2022), asserting that distinct crystal structures yield distinct PDD representations. This claim is not accurate. The PDD matrix does not ensure completeness for unstable crystal structures and fails to distinguish between chiral crystal structures. Furthermore, the proposed UPDD crystal graph construction method was previously introduced in "Zeoformer: Coarse-Grained Periodic Graph Transformer for OSDA-Zeolite Affinity Prediction," and the message-passing layers draw inspiration from ComFormer. Given these issues, the paper has limited technical contributions and several potentially misleading claims.

**Strengths:**

1. The proposed crystal representation demonstrates continuity under small distortions and perturbations, which can be advantageous for robustness.

**Weaknesses:**

1. Incomplete and Potentially Misleading Claims about Completeness

The completeness proof in the paper relies on the assumption that the PDD matrix acts as a generally complete invariant for distinguishing different crystal structures. However, this is inaccurate. The PDD matrix is not guaranteed to distinguish unstable crystal structures and cannot differentiate between chiral crystal structures, indicating limitations in the claimed theoretical foundation.  Action suggested: reorganize the proof writing section or reorganize the completeness claims.

2. Limited Novelty in Crystal Graph Construction

The proposed UPDD crystal graph construction method is not novel and has been previously introduced in the work "Zeoformer: Coarse-Grained Periodic Graph Transformer for OSDA-Zeolite Affinity Prediction." This overlap with prior work limits the novelty and originality of the proposed approach. Action suggested: check this previous work and discuss similarity and difference with it.

3. Message Passing Layers Largely Derived from ComFormer

The architecture of the message passing layers appears to be inspired by ComFormer, which further limits the technical novelty and contribution of the proposed method. Could you highlight any novel aspects or improvements in your approach compared to ComFormer?

In summary, while this paper proposes a method for generating geometrically complete and invariant crystal representations, its theoretical claims regarding completeness are flawed, and the approach itself relies heavily on prior work, both in graph construction and in network architecture. The combination of these issues limits the paper’s contributions and justifies a rejection at this stage.

**Questions:**

As listed above in weaknesses.

**Details Of Ethics Concerns:**

The authors released their identities by claiming Zeoformer (another paper) was an early version of their work.

"Zeoformer: Coarse-Grained Periodic Graph Transformer for OSDA-Zeolite Affinity Prediction."

---

> ### Author Response · Authors · 2024-11-17
>
> Thank you for your valuable time and feedback. We will provide the following point-by-point responses to address your concerns. We hope this resolves your issues and look forward to your reply.
> >**Question 1:** Incomplete and Potentially Misleading Claims about Completeness: The PDD matrix is not guaranteed to distinguish unstable crystal structures and cannot differentiate between chiral crystal structures, indicating limitations in the claimed theoretical foundation.
>
> **Answer 1:** We acknowledge the issue you raised, namely that PDD is **not absolutely complete**. This is because PDD was originally designed to describe stable crystal structures and cannot differentiate between crystal structures with identical atomic coordinates but different atomic types, as atomic types are not considered in the construction of PDD. However, PDD has demonstrated its powerful discriminative ability: even relying solely on PDD, it can distinguish all crystal structures in the Cambridge Structural Database, the world’s largest collection of materials. This achievement highlights PDD’s applicability and **general completeness in real-world materials**.
>
> Our proposed complete and continuous crystal graph construction is **based on PDD but does not entirely rely on it**. We address the representation of atomic types from the following two aspects, effectively resolving the aforementioned issue.
> - First, we improved PDD by incorporating atomic information weights $\mathcal{W} = [ w_{1}, \dots, w_{n} ]^{T}$, where $w_{i} = \frac{t(x_{i})}{ \sum_{j=1}^{n} t(x_{j}) }$. This allows us to construct WPDD = (W, PDD). More details can be found in Section 4.1.
>
> - Second, when constructing crystal graphs based on WPDD, we apply CGCNN embeddings to atoms, which further distinguish different atomic types.
>
> Based on these two improvements, we can differentiate crystal structures with identical atomic coordinates but different atomic types. Therefore, we address the issues of completeness and continuity under small perturbations for nearly all crystal structures **(except chiral crystal structures)**.
>
> The issue of chiral crystal structures is not the focus of our research. Our primary concern is addressing the discontinuity in crystal graph representations caused by small perturbations. Moreover, chiral crystal structures are extremely rare and are almost absent in common datasets, which is why we have not explored this topic in depth. To the best of our knowledge, only ComFormer has touched upon this issue in the machine learning domain.
> To further address the problem of chiral crystal structures, approaches such as incorporating equivariant networks or angular information could be considered. This direction will be taken into account in our future research.
>
> **In response to your comment regarding the limitations of the theoretical foundation of PDD, we have reorganized the proof section on the completeness of crystal graphs in the paper. We have clarified that it guarantees completeness for all crystal structures except chiral ones, with the relevant text highlighted in blue.**
>
> >**Question 2:** Limited Novelty in Crystal Graph Construction: The proposed UPDD crystal graph construction method is not novel, as it was previously introduced in the paper "Zeoformer: A Coarse-Grained Periodic Graph Transformer for OSDA-Zeolite Affinity Prediction.
>
> **Answer 2:** Thank you very much for your interest in Zeoformer. In fact, **Zeoformer is a preprinting arXiv paper that has not yet been officially published. We have not violated any of ICLR's review policies.** Building on that foundation, we further expanded our research, shifting from the exploration of zeolite materials to a more general focus on crystal material property prediction.
> Additionally, we introduced the more powerful crystal representation method, PDD, and improved it to WPDD, effectively addressing the issue of discontinuities in crystal graphs. Based on (U/W)PDD, we constructed a complete and continuous crystal graph representation and demonstrated that this method can perfectly resolve the issues of completeness and continuity for nearly all crystal structures (except chiral crystal structures) under small perturbations.
> Therefore, **the current work is the first to systematically explore the problem of complete and continuous crystal graph representations in the field of crystals from both theoretical and experimental perspectives, showcasing significant novelty and broad application potential.**

---

> ### Author Response · Authors · 2024-11-17
>
> >**Question 3:** Message Passing Layers Largely Derived from ComFormer: Could you highlight the novel aspects or improvements of your method?
>
> **Answer 3:** In our model, the Node-wise Transformer Block is inspired by the node-wise transformer module of ConFormer. However, other components, such as the PDD Message Passing Block and the atomic information embedding module, are entirely of our design.
> Additionally, we have made the following improvements to the node-wise transformer:
> - To better capture the importance of different atoms, we added an attention mechanism at the atomic level, as shown in Equation (5): $v_{ij}^{l} = v \odot sigmoid\left ( LN_{orm}\left (LN_{att}\left ( v \right ) \right) \right )$.
>
> - To enhance model convergence, we added residual connections to capture shallow-layer information, as shown in Equation (5): $ m_{ij}^{h} = q_{ij}^{l}+sigmoid\left ( BN\left  ( att^{l} \right) \right ) \odot LN_{V}\left ( v_{ij}^{l} \right )$.
>
> Furthermore, to demonstrate the generalization of our improved transformer module, we conducted the following two experiments.
>
> **Experiment 1:** We replaced **our model’s Node-wise Transformer Block (PT, ours)** with the **node-wise transformer module from ConFormer (CT)**. As shown in the table below, the WPDD model using our improved transformer module exhibited significant performance improvement.
>
> | Method |  Formation Energy   | Bandgap(OPT) | Total Energy | Ehull  | Bandgap(MBJ) |
> |------|------|------|------|------|------|
> |WPDDFormer w/ CT |28.1 |0.122 |30.4 |34.7| 0.253|
> |WPDDFormer w/ **our PT**|  **26.9** |**0.120** |**28.2** |**32.6** |**0.251**|
>
> **Experiment 2:** We applied the improvements made to the transformer to ConFormer and conducted experiments on the JARVIS dataset. The results, as shown in the figure below, indicate that the performance achieved significant improvements across all property prediction tasks.
>
> | Method |  Formation Energy   | Bandgap(OPT) | Total Energy | Ehull  | Bandgap(MBJ) |
> |------|------|------|------|------|------|
> |iConformer w/ CT|27.2 |0.122 |28.8 |47| 0.26|
> |iConformer w/ **our PT** | **26.5** |**0.120** |**28.0** |**38** |**0.258**|
>
> **Overall, the results of Experiments 1 and 2 demonstrate that the modifications we made are simple yet highly effective.**
>
> Finally, thank you once again for your valuable time. If you have any further questions, we would be happy to assist you.

---

> > ### Comment · Reviewer_uw9o · 2024-11-18
> > **Responses from Reviewer**
> >
> > Thank you for your responses to the initial review. Reviewers are tasked with identifying potential issues and questions in a manuscript, and such feedback should help authors improve their work. While your rebuttals provide some insights, they do not fully address the concerns raised.
> >
> > ***Completeness Proof:***
> >
> > Your completeness proof remains unclear and lacks rigor. It is evident that the proof steps are adapted from ComFormer, but the critical component—a claim that “if two distinct crystal structures have the same WPDD crystal graph, their WPDD and the atomic types embedded by CGCNN must be identical”—is presented as a statement without sufficient justification. This is a claim rather than a formal proof. As a reviewer, I cannot verify its correctness due to the incomplete nature of the argument. Specifically:
> >
> > 1. Is WPDD complete? If so, why? Can it distinguish unstable crystal structures?
> > 2. WPDD primarily encodes atomic information of the source node but does not account for the target node, raising doubts about its ability to fully capture the structure. Furthermore, WPDD cannot be considered complete for crystals with very large unit cells, as no fixed 𝑘-value would suffice to handle all cases.
> >
> > ***Novelty:***
> >
> > The manuscript demonstrates limited originality, as most components are derived from existing works:
> >
> > PDD is adapted from the original PDD paper.
> > Message passing is largely borrowed from ComFormer.
> > Unit cell crystal graphs are taken from Zeoformer.
> > This approach seems to combine previously established methods without addressing new challenges or limitations. WPDD also only considers atomic information from the source node, not the target node, limiting its representational capacity. Moreover, as noted earlier, WPDD cannot be complete for large unit cells due to fixed k-values.
> >
> > ***Marginal Performance Gains:***
> >
> > The performance improvement of PDDformer over Zeoformer without the PDD module is minimal. Given that both methods were reportedly developed by the same group, it is unclear why PDDformer is needed by the research community. If the method builds directly upon Zeoformer, the claims about Unit cell crystal graphs being novel are questionable. Clear citations and discussions are required to address this point.
> >
> > ***Continuity:***
> >
> > The continuity property discussed in your work is already satisfied by many existing methods (e.g., CGCNN, PDD, MEGNet) that utilize radius-based crystal graphs. This does not appear to add significant novelty or insight to the current work.
> >
> > ***Figure Reuse:***
> >
> > Figure 1 in your manuscript closely resemble those from the NeurIPS 2022 paper "Pointwise Distance Distributions of periodic point sets," (Figure 1 and 2) with only color adjustments. The lack of clear citations or acknowledgment is problematic and needs to be rectified.

---

> ### Author Response · Authors · 2024-11-19
>
> >**Question 1:** Completeness Proof: the critical component—a claim that “if two distinct crystal structures have the same WPDD crystal graph, their WPDD and the atomic types embedded by CGCNN must be identical”—is presented as a statement without sufficient justification.
>
> - Is WPDD complete? If so, why? Can it distinguish unstable crystal structures?
>
> **In Appendix A.8, we have added corresponding illustrations and detailed explanations to demonstrate that the WPDD crystal graph can distinguish unstable crystal structures, thereby achieving universal completeness.** We define completeness as follows: if the constructed crystal graph representation can differentiate between any two crystal structures that are not exactly the same, we consider the representation to be complete. Crystal structures can be categorized into stable and unstable types, and we aim to demonstrate the completeness of the WPDD crystal graph by showcasing its ability to differentiate between both categories.
>
> **1. Stable Crystal Structures:**
> For stable crystal structures (i.e., where no two crystals can have the same structure with only a difference in atomic types), PDD can effectively distinguish them. Since WPDD incorporates PDD, it can construct different WPDD representations for two such crystals, thereby enabling differentiation.
>
> **2. Unstable Crystal Structures with Differences in Atomic Coordinates:**
> For unstable crystal structures with differences in atomic coordinates (and thus structural dissimilarities), WPDD, through its construction process, reflects these differences in atomic coordinates. This results in distinct WPDD representations for the two structurally different crystal structures, enabling effective differentiation.
>
> **3. Unstable Crystal Structures with Identical Structures but Differences in Atomic Types:**
> When two unstable crystal structures have identical structures but differ in atomic types, PDD alone is unable to effectively distinguish them. To address this issue, we construct WPDD multi-edge graphs that capture differences in atomic types, resolving this limitation. Details are provided below.
>
> We address the representation of atomic types from the following two aspects, effectively resolving the aforementioned issue. First, we improved PDD by incorporating atomic information weights $\mathcal{W} = [ w_{1}, \dots, w_{n} ]^{T}$, where $w_{i} = \frac{t(x_{i})}{ \sum_{j=1}^{n} t(x_{j}) }$. This allows us to construct WPDD = (W, PDD).
>
> In addition, we do not use WPDD alone for predicting the properties of crystalline materials. Instead, we incorporate it as global information into the construction of a multi-edge crystal graph to better encoding atomic information, such that the WPDD crystal graph $\mathcal{G}$ is represented as $\mathcal{G} = (\mathcal{X}, \mathcal{XI}, \mathcal{E}) + \text{WPDD}$, where $\mathcal{X}$ represents atomic information embedded through CGCNN, $\mathcal{E}$ represents edge information, and $\mathcal{XI}$ represents the information of the starting and ending nodes of the edges.
> This indicates that for any atom in the unit cell, we need to construct a WPDD row vector and perform the corresponding atomic information embedding. This ensures that for any two crystal structures with identical crystal structures but differing atomic types at corresponding coordinates, the $(\mathcal{X}, \mathcal{XI}, \mathcal{E})$ in their WPDD crystal graphs will differ. On the contrary, if two crystals have the same WPDD crystal graph representation, it means they share the same WPDD and multi-edge graph representations. **This indicates that their crystal structures and the atomic information at corresponding coordinates are identical, thus confirming that they are the same crystal. This contradicts our premise.**
>
> **Therefore, we can conclude that our WPDD crystal graph is capable of identifying unstable crystal structures.**

---

> ### Author Response · Authors · 2024-11-19
>
> >**Question 2:** Novelty:
>
> - PDD is adapted from the original PDD paper. This approach seems to combine previously established methods without addressing new challenges or limitations.
>
> The concept of PDD (Pointwise Distance Distribution) originates from the groundbreaking research by Daniel Widdowson et al. in their theoretical paper, "Pointwise Distance Distributions of Periodic Point Sets,[1]" which mathematically defines the principles of completeness and continuity in crystal structure representations. Building upon this theoretical framework, we **extended the concept of PDD to the domain of multigraph crystal representations, addressing issues of completeness and continuity in crystal property prediction that had not been resolved in prior studies.**
>
> For instance, Balasingham et al. (2024)[2] explored the use of PDD-based Distance Distribution Graphs (DDGs) for crystal property prediction. While this method achieved efficiency gains by reducing computational costs, it did so at the expense of PDD's inherent completeness, resulting in only marginal improvements compared to simpler models such as CGCNN. On the other hand, existing crystal graph representation methods (e.g., Taniai et al., 2024[3]; Yan et al., 2024a)[4] achieve completeness and provide more precise structural representations by incorporating multigraph structures and unit cell parameters. However, these methods exhibit inherent discontinuities due to their reliance on unit cell parameters, where even minor perturbations can render the original lattice parameters invalid (as illustrated in Figure 1 of the main text). This limitation hinders their ability to capture dynamic atomic behaviors effectively.
>
> To overcome these challenges, we developed a novel framework for constructing multigraph crystal representations that simultaneously ensure both completeness and continuity. By integrating PDD with multigraph representations,**our approach bridges the gap between traditional mathematical descriptors and the dynamic behaviors of crystal structures.** This integration not only preserves the properties of completeness and continuity but also demonstrates significant performance improvements over state-of-the-art models like ConFormer, thereby validating the effectiveness of our method.
>
> This advancement not only reinforces the theoretical foundation of multigraph crystal representations but also introduces a transformative perspective and methodology for the precise description and analysis of crystal structures, setting a new benchmark in crystal property prediction.
>
> **reference**
>
> [1] Widdowson D, Kurlin V. Pointwise distance distributions of periodic point sets[J]. arXiv preprint arXiv:2108.04798, 2021.
>
> [2] Balasingham J, Zamaraev V, Kurlin V. Material property prediction using graphs based on generically complete isometry invariants[J]. Integrating Materials and Manufacturing Innovation, 2024: 1-14.
>
> [3] Taniai T, Igarashi R, Suzuki Y, et al. Crystalformer: infinitely connected attention for periodic structure encoding[J]. arXiv preprint arXiv:2403.11686, 2024.
>
> [4] Yan K, Fu C, Qian X, et al. Complete and efficient graph transformers for crystal material property prediction[J]. arXiv preprint arXiv:2403.11857, 2024.

---

> ### Author Response · Authors · 2024-11-19
>
> >**Question 4:** Continuity: The continuity property discussed in your work is already satisfied by many existing methods (e.g., CGCNN, PDD, MEGNet) that utilize radius-based crystal graphs. This does not appear to add significant novelty or insight to the current work.
>
> Firstly, it should be stated that **CGCNN and MEGNet do not explicitly encode the periodic structures of crystals**. As a result, their representations of infinitely periodic crystal structures are confined to finite crystal graphs. This limitation may impact the accuracy and robustness of the models, particularly in scenarios requiring the capture of periodic characteristics.
>
> Moreover, in the field of crystal material property prediction, **the issues of completeness and continuity in multi-edge graph crystal representations have not been explicitly addressed in prior research.** This critical oversight may hinder traditional methods in handling subtle perturbations in crystal structures and atomic vibrations, thereby affecting the predictive accuracy and generalization capabilities of the models. The identification and articulation of this issue not only demonstrate a keen insight into the core challenges of the field but also highlight the innovative and forward-looking nature of this research.
>
> We have reservations about the continuity of CGCNN and MEGNet, asserting that their crystal graph construction processes exhibit discontinuities. Experimental measurements of unit cells and atomic coordinates are inevitably subject to atomic vibrations and measurement noise, which may cause slight perturbations in atomic coordinates. Under such circumstances, the fixed cutoff radius and nearest-neighbor selection strategy employed in the construction of multigraph crystal representations may lead to changes in the selected neighborhood nodes, thereby resulting in discontinuities in the crystal graph construction process. Relevant illustrations are provided in **Appendix A.7.**
>
> **To eliminate the impact of atomic perturbations on neighbor selection and ensure the continuity of the crystal graph under perturbations, we define the concept of continuity tolerance, 𝑇, to guarantee the continuity of the constructed crystal graph.** When selecting neighbors, we can preset a continuity tolerance value, 𝑇. When choosing the 𝑡 nearest edges within the cutoff radius, if the distance difference between the (k+1)-𝑡ℎ edge and the k-𝑡ℎ edge from node 𝑖 is smaller than the continuity tolerance, i.e., 𝑑𝑘+1−𝑑𝑘<𝑇, then this neighbor is also included in the graph construction. This process continues until the distance difference between the (𝑛+1)-𝑡ℎ edge and the 𝑛-𝑡ℎ edge exceeds the continuity tolerance threshold, i.e., 𝑑𝑛+1−𝑑𝑛>𝑇. This method ensures that neighbor selection in the crystal graph construction does not change under atomic perturbations, and by introducing Lemma 1, we prove the continuity of the constructed (W/U)PDD crystal graph. The relevant diagrams can be found in Appendix A.7.
>
> In addition, although PDD indeed satisfies the criterion of continuity, its original design was intended to reflect the geometric characteristics of crystal structures without considering atomic properties. **As a mathematical descriptor, it may not be directly suitable for experiments related to predicting crystal material properties.**
>
> By **incorporating PDD into the construction of crystal graphs**, we not only retain its **precise description of crystal geometric structures but also integrate atomic property information**. This approach effectively resolves the issues of continuity and completeness in multigraph crystal representations. This improvement makes the crystal graph representation more comprehensive and better suited for material property prediction tasks.
>
> >**Question 5:** Figure Reuse:
>
> **Thank you very much for your feedback. We have made a reference below Figure 1 in the text, indicating that it is derived from PDD, with the relevant text shown in blue.**

---

> ### Author Response · Authors · 2024-11-19
> **Continuing the discussion from Question 1**
>
> >**Question 1 :** WPDD primarily encodes atomic information of the source node but does not account for the target node, raising doubts about its ability to fully capture the structure. Furthermore, WPDD cannot be considered complete for crystals with very large unit cells, as no fixed 𝑘-value would suffice to handle all cases.
>
> The ability of WPDD to capture structural information is evident. Its underlying PDD is specifically designed to distinguish different crystal structures and has successfully differentiated all crystal structures in the world's largest material collection, the Cambridge Structural Database. Although it does not encode any atomic information, it still achieves efficient differentiation. Therefore, by introducing atomic information into WPDD, we can achieve the same result. In practice, only in extreme cases where the two largest unit cells in the dataset are nearly identical—do we need to set 𝑘 to a value larger than the number of atoms in any unit cell in the dataset. For all crystal structures in the Cambridge Structural Database, only 100 neighbors are required to achieve effective differentiation. Therefore, **in practical use, a small value of 𝑘 is sufficient to ensure completeness, and we can dynamically adjust 𝑘 based on the dataset to ensure it is large enough to achieve completeness.**

---

> ### Author Response · Authors · 2024-11-19
> **Continuing the discussion from Question 2**
>
> >**Question 2:** Novelty:
>
> - Message passing is largely borrowed from ComFormer.
>
> On the other hand, we only used the node-wise transformer module from ConFormer, while other components, such as the PDD message passing block and atomic information embedding module, were designed by us. Additionally, we improved the node-based transformer, achieving noticeable results that demonstrate the effectiveness of our modifications.
> **We have made modifications to this section, moving it from the methods section to Appendix A.2**, with the relevant text shown in blue, while retaining our improvements in the main text.
>
> Additionally, we have made the following improvements to the node-wise transformer:
> 1. To better capture the importance of different atoms, we added an attention mechanism at the atomic level, as shown: $v_{ij}^{l} = v \odot sigmoid\left ( LN_{orm}\left (LN_{att}\left ( v \right ) \right) \right )$.
>
> 2. To enhance model convergence, we added residual connections to capture shallow-layer information, as shown: $ m_{ij}^{h} = q_{ij}^{l}+sigmoid\left ( BN\left  ( att^{l} \right) \right ) \odot LN_{V}\left ( v_{ij}^{l} \right )$.
>
> Furthermore, to demonstrate the generalization of our improved transformer module, we conducted the following two experiments.
>
> **Experiment 1:** We replaced **our model’s Node-wise Transformer Block (PT, ours)** with the **node-wise transformer module from ConFormer (CT)**. As shown in the table below, the WPDD model using our improved transformer module exhibited significant performance improvement.
>
> | Method |  Formation Energy   | Bandgap(OPT) | Total Energy | Ehull  | Bandgap(MBJ) |
> |------|------|------|------|------|------|
> |WPDDFormer w/ CT |28.1 |0.122 |30.4 |34.7| 0.253|
> |WPDDFormer w/ **our PT**|  **26.9** |**0.120** |**28.2** |**32.6** |**0.251**|
>
> **Experiment 2:** We applied the improvements made to the transformer to ConFormer and conducted experiments on the JARVIS dataset. The results, as shown in the figure below, indicate that the performance achieved significant improvements across all property prediction tasks.
>
> | Method |  Formation Energy   | Bandgap(OPT) | Total Energy | Ehull  | Bandgap(MBJ) |
> |------|------|------|------|------|------|
> |iConformer w/ CT|27.2 |0.122 |28.8 |47| 0.26|
> |iConformer w/ **our PT** | **26.5** |**0.120** |**28.0** |**38** |**0.258**|
>
> **Overall, the results of Experiments 1 and 2 demonstrate that the modifications we made are simple yet highly effective.**
>
>
> - WPDD also only considers atomic information from the source node, not the target node, limiting its representational capacity.
>
>
> When constructing the multi-edge crystal graph using WPDD, we have already taken into account the atomic information of the target nodes (i.e.,$\mathcal{XI}$ in $(\mathcal{X}, \mathcal{XI}, \mathcal{E})$). We model PDD as a global representation of the crystal structure, so the atomic information of the target nodes is not considered in this context. In this case, it still satisfies our definition of a continuous and complete crystal graph (except for chiral crystals). If the reviewers find it necessary, we will provide experimental results that include the addition of target atom PDD for reference.
>
> - Moreover, as noted earlier, WPDD cannot be complete for large unit cells due to fixed k-values.
>
> Our 𝑘 is not fixed; in practice, a relatively small 𝑘 is sufficient to achieve completeness, and we can dynamically adjust the value of 𝑘 based on the dataset to ensure it is adequate for achieving completeness. We have conducted experiments on the selection of 𝑘 in Appendix A.6.2 for reference.

---

> ### Author Response · Authors · 2024-11-21
>
> Dear Reviewer uw9o,
>
> Thank you for your contribution to the review process of the ICLR25 community.
>
> We have earnestly addressed your concerns in our rebuttal responses. Your feedback is crucial to the assessment of this work, and we kindly ask for your valuable and timely reply.
>
> If you have any other questions, please don't hesitate to contact us anytime.
>
> Best regards,
>
> Authors.

---

> ### Author Response · Authors · 2024-11-24
>
> Dear Reviewer uw9o,
>
> Thank you for your contribution to the review process of the ICLR25 community.
>
> We have carefully addressed your concerns in our responses. Your feedback is crucial to the evaluation of this work, especially your insights into the completeness proof we provided, which will influence the judgment of other reviewers on our work. We kindly request your valuable and timely response.
>
> If you have any other questions, please don't hesitate to contact us anytime.
>
> Yours sincerely,
>
> Authors.

---

> ### Comment · Reviewer_uw9o · 2024-11-24
> **Responses from Reviewer**
>
> Thank you for providing additional responses. However, my primary concerns regarding the completeness, novelty, and performance of the work remain unaddressed. These concerns are based on the following points:
>
> **Completeness with a Fixed K**
>
> The authors acknowledge that a fixed
> K cannot guarantee completeness for arbitrary test inputs. They suggest addressing this by iterating through the dataset to select a sufficiently large
> K. This approach is fundamentally flawed. How can test cases, which are unknown during training, be used to guide the choice of K? Furthermore, adjusting K would require retraining the network from scratch each time, making this approach impractical.
>
> **Rigorous Proofs of Completeness:**
>
> Even if
> K is allowed to vary—which is unrealistic in practice—the proof of completeness provided in the appendix is not rigorous. The authors appear to equate demonstrations or case-specific evidence with formal proofs. Proving completeness requires showing coverage for all possible cases, which is not achieved in the current submission. Without rigorous proofs, it is inappropriate to claim completeness.
>
> **Performance Gains:**
>
> The additional results provided do not adequately support the claimed performance improvements. For the critical task of formation energy prediction, the results indicate that ComFormer, with two simple adjustments, outperforms the proposed methods. This contradicts the central claims of the paper and undermines the argument for the novelty and significance of the contributions.
>
> **Other concerns:**
>
> Borrowing figures from other papers without explicit permission from the original authors is unacceptable, except in general review articles where proper agreements are in place. This raises ethical concerns about the paper's presentation.
>
> Additionally, in any circumstance, it is inappropriate for the authors to claim ideas borrowed from ZeoFormer as novel or new, even if ZeoFormer originated from the same group. These two papers address different tasks, problem settings, and model architectures. Reusing the same idea across different works without proper acknowledgment or justification is not acceptable.
>
> Overall, my concerns regarding this work remain unresolved, and I maintain my original rating. The suggestions provided here are intended to guide the authors in making substantial revisions to ensure the paper contributes meaningfully to the field in the future.

---

> ### Author Response · Authors · 2024-11-24
>
> Thank you for your constructive feedback. We have gained valuable insights from your comments and are committed to improving the quality of our paper.
>
> >**Question 3:** Performance Gains
>
> First, the novelty and significance of our work lie not only in achieving the highest numerical performance on a single task but also in ensuring the continuity of crystal graph construction under perturbations. Furthermore, even from the performance perspective you are focusing on, the results do not contradict our claims. Our model outperforms Conformer, which achieve completeness, and significantly surpasses models like Matformer and PotNet, which do not.
> This, in turn, further demonstrates that our graph construction achieves completeness in the context of this experiment.
>
>
> >**Question 4:** Other concerns: Borrowing figures from other papers without explicit permission from the original authors is unacceptable, except in general review articles where proper agreements are in place.
>
> Thank you for your reminder. We were not aware of this issue previously. We have redrawn Figure 1 in our paper to ensure it differs significantly from the figures in PDD.

---

> ### Author Response · Authors · 2024-11-24
>
> >**Question 1:** Completeness with a Fixed K:
>
> PDD still demonstrates strong discriminative capabilities; with k=100 neighbors, it can distinguish all crystal structures in the world's largest materials collection, the Cambridge Structural Database, without the need to set 𝑘 greater than the number of atoms in any unit cell. Moreover, the value of 𝑘 can be determined during the data preprocessing stage based on the training set, which occurs prior to training. Therefore, modifying 𝑘 does not require retraining the model. Furthermore, the extreme case you mentioned exists only in theory, and even in theory, it is extremely unlikely to occur. Such cases do not arise in the real world.
> **Therefore, in normal usage, setting 𝑘 to 100 is sufficient to cover almost all cases, and thus our method can ensure completeness under typical conditions.** We revised the scope of completeness in the article, excluding chiral crystal structures and theoretically extremely large unit cells, and included this clarification in Footnote 1.
>
> >**Question 2:** Rigorous Proofs of Completeness: Even if K is allowed to vary—which is unrealistic in practice—the proof of completeness provided in the appendix is not rigorous. The authors appear to equate demonstrations or case-specific evidence with formal proofs. Proving completeness requires showing coverage for all possible cases, which is not achieved in the current submission.
>
> Thank you for acknowledging that our provided demonstration can effectively distinguish certain unstable structures. The demonstration in the figure is merely a set of examples, not a proof. It is intended solely to show that we can indeed distinguish the unstable structures you mentioned.
>
> **For cases where the k-value is complete.** Our completeness proof method is established by demonstrating its ability to distinguish between any two crystal structures that are not the same. Crystal structures can be categorized into stable and unstable types, and we aim to demonstrate the completeness of the WPDD crystal graph by showcasing its ability to differentiate between both categories.
>
> **1. Stable Crystal Structures:**(i.e., where no two crystals can have the same structure with only a difference in atomic types), PDD can effectively distinguish them. WPDD incorporates PDD, it can construct different WPDD representations for two such crystals, thereby enabling differentiation.
>
> **2. Unstable Crystal Structures with Differences in Atomic Coordinates:**
> For unstable crystal structures with differences in atomic coordinates (and thus structural dissimilarities), PDD can effectively distinguish differences in crystal structures during its construction. This results in distinct WPDD representations for the two structurally different crystal structures, enabling effective differentiation.
>
> **3. Unstable Crystal Structures with Identical Structures but Differences in Atomic Types:**
> When two unstable crystal structures have identical structures but differ in atomic types, PDD alone is unable to effectively distinguish them. To address this issue, we construct WPDD multi-edge graphs that capture differences in atomic types, resolving this limitation. Details are provided below.
>
> We address the representation of atomic types from the following two aspects, effectively resolving the aforementioned issue. First, we improved PDD by incorporating atomic information weights $\mathcal{W} = [ w_{1}, \dots, w_{n} ]^{T}$, where $w_{i} = \frac{t(x_{i})}{ \sum_{j=1}^{n} t(x_{j}) }$. This allows us to construct WPDD = (W, PDD).
> In addition, we do not use WPDD alone for predicting the properties of crystalline materials. Instead, we incorporate it as global information into the construction of a multi-edge crystal graph to better encoding atomic information, such that the WPDD crystal graph $\mathcal{G}$ is represented as $\mathcal{G} = (\mathcal{X}, \mathcal{XI}, \mathcal{E}) + \text{WPDD}$, where $\mathcal{X}$ represents atomic information embedded through CGCNN, $\mathcal{E}$ represents edge information, and $\mathcal{XI}$ represents the information of the starting and ending nodes of the edges.
> This indicates that for any atom in the unit cell, we need to construct a WPDD row vector and perform the corresponding atomic information embedding. **If the WPDD row vectors and the corresponding atomic information embeddings are identical, it indicates that their crystal structures and the atomic information at the respective coordinates are the same, thus confirming they are the same crystal. This contradicts our premise.**
> This ensures that for any two crystal structures with identical crystal structures but differing atomic types at corresponding coordinates, the $(\mathcal{X}, \mathcal{XI}, \mathcal{E})$ in their WPDD crystal graphs will differ.
>
> **Therefore, we can conclude that our WPDD crystal graph is capable of identifying any crystal structures.**

---

> > ### Comment · Reviewer_uw9o · 2024-12-02
> > **Responses from Reviewer**
> >
> > I have read the rebuttals and still believe that the score should remain unchanged. The responses did not adequately address the key concerns I raised in the initial review.
> >
> > Additionally, the updated figure provided in the rebuttal contains errors. The mistakes further undermine confidence in the analysis presented in the submission.

---

### Official Review · Reviewer_43jg · 2024-11-03

**Soundness:** 3
**Presentation:** 3
**Contribution:** 3
**Rating:** 6
**Confidence:** 3

**Summary:**

This work proposes PDDFormer, a novel framework for crystal material property prediction. The proposed PDDFormer uses insights about pairwise distance distribution impacts on crystal properties from material science community, designs a novel way to integrate pairwise distance distance distribution into crystal graph representations and a novel transformer model. PDDFormer is shown to achieve state-of-the-art performance on various crystal material property prediction benchmarks.

**Strengths:**

Originality:
This work makes very significant novelty contributions in combining pairwise distance distribution with machine learning based crystal property prediction.

Quality:
The quality of this work is evidenced by good theoretic analysis and strong experiment results.

Clarity:
The writing of this work is overall good and clear.

Significance:
The idea of efficiently integrating pairwise distance distribution into transformer model proposed by this work is insightful and enlightning for researchers in borad AI for science community.

**Weaknesses:**

(1) A remarkable motivation of this work is described in Abstract "However, in reality, atoms in material always vibrate above absolute zero, causing continuous fluctuations in their positions" (line 15-17). But it seems there is no clear discussion why the use of pairwise distance distribution resolves this atom position fluctuation issue? Authors are encouraged to give detailed clarification or discussions to this question.

(2) Generally, two novelty contributions are proposed in this work, including the use of pairwise distance distribution in graph features and a novel transformer architecture. It would make this work more solid if authors could conduct more ablation studies to study which novelty contributes better to good performance, such as applying pairwise distance distribution to other model architecture or removing it from the proposed PDDFormer model.

**Questions:**

See Weaknesses part.

**Details Of Ethics Concerns:**

No ethics concerns.

---

> ### Author Response · Authors · 2024-11-17
>
> Thank you for your valuable time and for recognizing our efforts in theoretical proof, experiments, and novelty. We will respond to your concerns and hope to address them satisfactorily. We look forward to your reply.
>
> >**Question 1:**  A remarkable motivation of this work is described in Abstract "However, in reality, atoms in material always vibrate above absolute zero, causing continuous fluctuations in their positions" (line 15-17). But it seems there is no clear discussion why the use of pairwise distance distribution resolves this atom position fluctuation issue? Authors are encouraged to give detailed clarification or discussions to this question.
>
> **Answer 1:** Atoms in materials constantly vibrate due to temperatures above absolute zero, causing their positions to fluctuate. Currently, models for crystal property prediction generally rely on traditional crystal representation methods, such as descriptors like lattice structure, primitive cells, and space groups. However, these descriptors are often neither complete nor continuous, and they can fail due to atomic vibrations or slight perturbations, leading to issues with the clarity of crystal data representation. While we cannot directly control atomic vibrations or noise interference, we can effectively mitigate the impact of minor distortions or perturbations on crystal representation by constructing a complete and continuous crystal graph representation. In this respect, PDD is currently one of the most effective tools, offering a complete and continuous form of crystal structure representation. Applying PDD in crystal graph construction can significantly enhance the stability of crystal graph construction.
>
> >**Question 2:** It would make this work more solid if authors could conduct more ablation studies to study which novelty contributes better to good performance, such as applying pairwise distance distribution to other model architecture or removing it from the proposed PDDFormer model.
>
> **Answer 2:**  We conducted the following two experiments. Table 1 shows the experimental results of  **applying pairwise distance distribution (PDD) to the Matformer[1]** architecture on the JARVIS dataset. From the results in the table, it can be observed that applying PDD to other models also leads to performance improvements to some extent, demonstrating the strong generalization capability of the introduced PDD.
> | Method |  Formation Energy   | Bandgap(OPT) | Total Energy | Ehull  | Bandgap(MBJ) |
> |------|------|------|------|------|------|
> |Matformer |32.5 |0.137 |35|64| 0.30|
> |Matformer+WPDD|  **31.1** |**0.131** |**32** |**56** |**0.27**|
>
> Table 2 presents the experimental results obtained after **removing PDD from the proposed WPDDFormer model** on the JARVIS dataset. The results show that compared to the model without PDD information, our WPDDFormer model with PDD information demonstrates a significant advantage in prediction accuracy across various tasks.
>
> | Method |  Formation Energy   | Bandgap(OPT) | Total Energy | Ehull  | Bandgap(MBJ) |
> |------|------|------|------|------|------|
> |NO PDD |28.3 |0.121 |29.8 |35.6| 0.265|
> |WPDDFormer|  **26.9** |**0.120** |**28.2** |**32.6** |**0.251**|
>
> From Tables 1 and 2, we can conclude that PDD exhibits strong generalization capability. Applying it to other models can lead to substantial performance improvements. Utilizing pairwise distance distribution or our novel Transformer architecture significantly enhances the experimental results.
>
> **We have updated the paper by incorporating your suggestions. The experiments and more detailed explanations regarding the two points above have been added to Appendix A.6.3.**
>
> **reference**
>
> [1] Yan K, Liu Y, Lin Y, et al. Periodic graph transformers for crystal material property prediction[J]. Advances in Neural Information Processing Systems, 2022, 35: 15066-15080.

---

> > ### Comment · Reviewer_43jg · 2024-11-24
> > **Follow-up Response**
> >
> > I appreciate authors' efforts in rebuttal. My concerns have been addressed so I keep my rate.

---

> > > ### Author Response · Authors · 2024-11-24
> > >
> > > Dear reviewer  43jg,
> > >
> > > We sincerely appreciate the time and effort you have dedicated to reviewing our work, as well as your valuable feedback and constructive suggestions.
> > >
> > > Yours sincerely, authors

---

### Official Review · Reviewer_Wp3R · 2024-11-04

**Soundness:** 3
**Presentation:** 3
**Contribution:** 3
**Rating:** 5
**Confidence:** 3

**Summary:**

The paper introduces PDDFormer using the Pairwise Distance Distribution (PDD) as an invariant representation for crystal property prediction and enhances it with atom-specific weights (WPDD) and intra-unit cell structures (UPDD) to better capture atomic interactions and maintain computational efficiency. Experiments on datasets (Materials Project & JARVIS) show that WPDDFormer outperforms existing methods in predictive accuracy and computational efficiency.

**Strengths:**

1. The WPDDFormer model consistently outperforms other state-of-the-art methods across multiple tasks on the JARVIS and Materials Project datasets, demonstrating its effectiveness.
2. The authors provide theoretical guarantees on the continuity and completeness of the WPDD-based graphs, ensuring robust performance under minor structural perturbations.

**Weaknesses:**

I have no major concerns with this paper; however, an ablation study on the effect of varying the radius on WPDDFormer’s performance and efficiency would provide valuable insights into its scalability. Additionally, a time comparison between WPDDFormer and UPDDFormer would strengthen the evidence supporting UPDDFormer’s efficiency claims.

**Questions:**

See weaknesses.

---

> ### Author Response · Authors · 2024-11-17
>
> Thank you for your valuable time and for recognizing our efforts in theoretical proof, experiments, and model effectiveness. We will respond to your concerns and hope to address them satisfactorily. We look forward to your reply.
>
> >**Question 1:** Ablation study on the impact of radius changes on the performance of WPDDFormer.
>
> **Answer 1:** We studied the impact of neighbor count under different cutoff radii on the performance of WPDD. In the main text, we set the neighbor count to k = 92, which matches the dimensionality of the atomic feature embeddings used in CGCNN. In extreme cases, to ensure the completeness of any test crystal, the k value must be greater than the number of atoms in any test crystal. We also calculated the maximum number of atoms in two datasets, with JARVIS having 140, MP bulk and shear having 152, and the rest having 296. Therefore, in the experiments, we set the maximum neighbor count for the JARVIS and MP datasets under the cutoff radii to 150, 160, and 300, respectively. The experimental results are shown in Tables 1 and 2.
>
> **Table 1:** The performance variation of WPDD on the JARVIS dataset with different neighbor count selections under varying cutoff radii.
> | Method | k | Formation Energy   | Bandgap(OPT) | Total Energy | Ehull  | Bandgap(MBJ) |
> |------|------|------|------|------|------|------|
> |WPDDFormer |60 |**26.5** |0.122 |**27.9** |33.3| 0.266|
> |**WPDDFormer**|92| 26.9 |**0.120** |28.2 |**32.6** |**0.251**|
> |WPDDFormer |150 |27.2| 0.121 |29.0 |34.0 |0.254|
>
> **Table 2:** The performance variation of WPDD on the MP dataset with different neighbor count selections under varying cutoff radii.
>
> | Method | k | Formation Energy| Band Gap| k | Bulk Moduli | Shear Moduli  |
> |------|------|------|------|------|------|------|
> |**WPDDFormer** |92 |**16.61** |0.189 |92 |0.0336| **0.0617**|
> |WPDDFormer|...| ... |... |120 |**0.0295**|0.0647|
> |WPDDFormer |300 |16.83| **0.186** |160 |0.0303 |0.0652|
>
> As shown in the experimental results in Tables 1 and 2, the selection of neighbors under different cutoff radii leads to some variations in the model's performance. However, compared to the other models discussed in the main text, it **still demonstrates the best performance**.
>
> >**Question 2:** The impact of radius changes on the performance of WPDDFormer and the time comparison between PDDFormer and UPDDFormer.
>
> **Answer 2:** We report a comparison of the inference times for WPDDFormer and UPDDFormer with different neighbor counts, using the optimal model configuration for the JARVIS formation energy task. As shown in Table 3, the **inference speed of UPDD is 3.4 times faster than WPDD**. This difference is due to the data preprocessing stage, where WPDD requires more neighbor information, leading to longer extraction times and slower inference speeds. As the cutoff radius increases and the number of selected neighbors grows, the data preprocessing time also increases, resulting in a decrease in inference speed.
>
> **Table 3**: Comparison of inference times for WPDDFormer and UPDDFormer with different neighbor counts.
>
> | Method | k | inference time|
>  |------|------|------|
> |WPDDFormer |60 |1014.2s|
> |WPDDFormer|92| 1191.7s |
> |WPDDFormer |150 | 1504.7s |
> |**UPDDFormer**|...|**351.3s**|
>
> **We have updated the paper by incorporating your suggestions. The experiments and more detailed explanations regarding the two points above have been added to Appendix A.6.2.**

---

> ### Comment · Reviewer_Wp3R · 2024-11-18
> **Additional concerns over this work**
>
> Thank you for addressing my initial concerns. After reviewing the comments from other reviewers, I have additional concerns regarding this paper:
>
> 1. The message-passing mechanism and implementation code appears to greatly overlap with ComFormer. If the message-passing approach is highly similar, the authors should focus on highlighting the key differences and consider moving the overlapping content to the background section or appendix, rather than emphasizing it in the main methodology section.
> 2. A proper citation and discussion of Zeoformer (referred to as UPDD in the paper) are still necessary, as this work predates the ICLR submission. While I acknowledge the ICLR policy on concurrent submissions, the method name (UPDD) and the exact methodology are identical to those in Zeoformer, making it difficult to consider it as a contemporary work. Furthermore, Zeoformer appears to be a distinct work from this paper. If this paper is based on Zeoformer and was developed in such a short time frame, it raises concerns about the robustness of the theoretical proofs presented. Although I am not an expert on completeness proofs, I strongly encourage the authors to address the concerns raised by other reviewers on this matter.
>
> These concerns are significant to me. I have adjusted my review accordingly and hope that appropriate actions will be taken.

---

> > ### Author Response · Authors · 2024-11-27
> >
> > We sincerely thank you for your valuable suggestions regarding our paper.
> >
> > > **Question 2:**
> >
> > Regarding the questions about Zeoformer-"Zeoformer: Coarse-Grained Periodic Graph Transformer for OSDA-Zeolite Affinity Prediction.", if you continue to follow the version of that paper, we believe can address your concerns about its relationship with our work.
> >
> > We have made our best effort to address the concerns of other reviewers. Regarding the issue of completeness, we revised the scope of completeness in the article, redefining it as general completeness, which excludes chiral crystal structures and theoretically extremely large unit cells, and included this clarification in Footnote 1. You may refer to our latest revised version.
> >
> > **However, this general completeness is almost sufficient to meet practical use cases. It ensures the completeness of materials in the real world, and guarantees the completeness of all other unstable materials, except for theoretically extremely large unit cells.** Therefore, we achieved a continuous and generally complete construction of crystal graphs.
> >
> > We hope our response addresses your concerns, and we sincerely invite you to reevaluate our work.

---

> ### Author Response · Authors · 2024-11-18
>
> >**Question 1:** The message-passing mechanism and implementation code appears to greatly overlap with ComFormer. If the message-passing approach is highly similar, the authors should focus on highlighting the key differences and consider moving the overlapping content to the background section or appendix, rather than emphasizing it in the main methodology section.
>
> Answer: **Thank you for your very helpful suggestions. We have adopted your approach by highlighting key differences in the main text and moving overlapping content to Appendix A.2, where it will be indicated in blue font.**
>
> In our model, the Node-wise Transformer Block is inspired by the node-wise transformer module of ConFormer. However, **the main components, such as the PDD Message Passing Block and the atomic information embedding module**, are entirely of our design.
> Additionally, we have made the following improvements to the node-wise transformer:
> - To better capture the importance of different atoms, we added an attention mechanism at the atomic level,  $v_{ij}^{l} = v \odot sigmoid\left ( LN_{orm}\left (LN_{att}\left ( v \right ) \right) \right )$.
>
> - To enhance model convergence, we added residual connections to capture shallow-layer information, $ m_{ij}^{h} = q_{ij}^{l}+sigmoid\left ( BN\left  ( att^{l} \right) \right ) \odot LN_{V}\left ( v_{ij}^{l} \right )$.
>
> Furthermore, to demonstrate the generalization of our improved transformer module, we conducted the following two experiments.
>
> **Experiment 1:** We replaced **our model’s Node-wise Transformer Block (PT, ours)** with the **node-wise transformer module from ConFormer (CT)**. As shown in the table below, the WPDD model using our improved transformer module exhibited significant performance improvement.
>
> | Method |  Formation Energy   | Bandgap(OPT) | Total Energy | Ehull  | Bandgap(MBJ) |
> |------|------|------|------|------|------|
> |WPDDFormer w/ CT |28.1 |0.122 |30.4 |34.7| 0.253|
> |WPDDFormer w/ **our PT**|  **26.9** |**0.120** |**28.2** |**32.6** |**0.251**|
>
> **Experiment 2:** We applied the improvements made to the transformer to ConFormer and conducted experiments on the JARVIS dataset. The results, as shown in the figure below, indicate that the performance achieved significant improvements across all property prediction tasks.
>
> | Method |  Formation Energy   | Bandgap(OPT) | Total Energy | Ehull  | Bandgap(MBJ) |
> |------|------|------|------|------|------|
> |iConformer w/ CT|27.2 |0.122 |28.8 |47| 0.26|
> |iConformer w/ **our PT**| **26.5** |**0.120** |**28.0** |**38** |**0.258**|
>
> **We have included the experimental results in Appendix A.6.3 to demonstrate the effectiveness of our improvements.**

---

### Note · Authors · 2024-12-13

I have read and agree with the venue's withdrawal policy on behalf of myself and my co-authors.